# Temporal inhibition of autophagy reveals segmental reversal of ageing with increased cancer risk

Liam D. Cassidy [1], Andrew R.J. Young[1], Christopher N.J. Young [2], Elizabeth J. Soilleux[3], Edward Fielder[4], Bettina M. Weigand[4,5,6], Anthony Lagnado[5,6], Rebecca Brais[7], Nicholas T. Ktistakis[8], Kimberley A. Wiggins[9], Katerina Pyrillou [9], Murray C.H. Clarke[9], Diana Jurk [5,6], Joao F. Passos[4,5,6] & Masashi Narita [1,10]*

Autophagy is an important cellular degradation pathway with a central role in metabolism as well as basic quality control, two processes inextricably linked to ageing. A decrease in autophagy is associated with increasing age, yet it is unknown if this is causal in the ageing process, and whether autophagy restoration can counteract these ageing effects. Here we demonstrate that systemic autophagy inhibition induces the premature acquisition of age-associated phenotypes and pathologies in mammals. Remarkably, autophagy restoration provides a near complete recovery of morbidity and a significant extension of lifespan; however, at the molecular level this rescue appears incomplete. Importantly autophagy-restored mice still succumb earlier due to an increase in spontaneous tumour formation. Thus, our data suggest that chronic autophagy inhibition confers an irreversible increase in cancer risk and uncovers a biphasic role of autophagy in cancer development being both tumour suppressive and oncogenic, sequentially.

---

[1] University of Cambridge, Cancer Research UK Cambridge Institute, Robinson Way, Cambridge CB2 0RE, UK. [2] Leicester School of Allied Health Sciences, Faculty of Health & Life Sciences, De Montfort University, Leicester LE1 5RR, UK. [3] Department of Pathology, University of Cambridge, Tennis Court Road, Cambridge CB2 1QP, UK. [4] Institute for Cell and Molecular Biosciences, Newcastle University Institute for Ageing, Newcastle University, Newcastle upon Tyne, UK. [5] Department of Physiology and Biomedical Engineering, Mayo Clinic, Rochester, MN, USA. [6] Robert and Arlene Kogod Center on Aging, Mayo Clinic, Rochester, MN, USA. [7] Department of Histopathology, Cambridge University Hospitals NHS Foundation Trust, Cambridge, UK. [8] Signalling Programme, Babraham Institute, Babraham, Cambridge, UK. [9] Division of Cardiovascular Medicine, Department of Medicine, University of Cambridge, Addenbrookes Hospital, Hills Road, Cambridge CB2 0QQ, UK. [10] Tokyo Tech World Research Hub Initiative (WRHI), Institute of Innovative Research, Tokyo Institute of Technology, Yokohama, Japan. *email: Masashi.Narita@cruk.cam.ac.uk

Physiological ageing is a complex and multifaceted process associated with the development of a wide array of degenerative disease states. While there is no accepted singular underlying mechanism of ageing, a combination of genetic, environmental and metabolic factors have been shown to alter the ageing process[1–3]. As such, lifestyle and pharmacological regimens have been proposed that may offer health- and or life-span benefits[4–6]. However, despite chronological ageing representing the greatest risk factor for pathological conditions as diverse as neurodegeneration, cancer, and cardiovascular disease, there is a paucity of genetic mammalian models that allow for dynamic modulation of key processes in mammalian ageing.

Autophagy is an evolutionarily conserved bulk cellular degradation system that functions to breakdown and recycle a wide array of cytoplasmic components from lipids, proteins and inclusion bodies, to whole organelles (e.g. mitochondria). Importantly a reduction in autophagic flux (the rate at which autophagosomes form and breakdown cellular contents) is associated with increasing age in mammals[7]. Evidence from lower organisms suggests that autophagy inhibition can negate the positive-effects of regimens that extend lifespan, such as calorie restriction, rapamycin supplementation, and mutations in insulin signalling pathways[8–10].

In mice, the constitutive promotion of autophagy throughout lifetime has been shown to extend health- and life-span in mammalian models[11,12]. These studies have provided hitherto missing evidence that autophagic flux can impact on mammalian longevity and supports the notion that the pharmacological promotion of autophagy may extend health-, and potentially life-span, in humans. However, whether a reduction in autophagy is sufficient to induce phenotypes associated with ageing, and whether these effects can be reversed by restoring autophagy has to date not been addressed. Considering that the therapeutic window for pharmacological intervention to counteract ageing, and age-related diseases, will be later in life (as opposed to from conception), after autophagic flux has declined, it is critical to understand how the temporal modulation (inhibition and restoration) of autophagy may impact on longevity and health.

To address these questions, we use two doxycycline (dox) inducible shRNA mouse models that target the essential autophagy gene Atg5 (Atg5i mice) to demonstrate that autophagy inhibition in young adult mice is able to drive the development of ageing-like phenotypes and reduce longevity. Importantly we confirm that the restoration of autophagy is associated with a substantial restoration of health- and life-span, however this recovery is incomplete. Notably the degree of recovery is segmental, being dependent on both the tissue and metric analysed. A striking consequence of this incomplete restoration is that autophagy restored mice succumb to spontaneous tumour formation earlier and at an increased frequency than control mice, a phenotype not observed during autophagy inhibition alone. As such our studies indicate that despite the significant benefit, autophagy reactivation may also promote tumorigenesis in advanced ageing context.

## Results

**Reduced lifespan in Atg5i mice.** Previously, we have reported the development of a highly efficient dox-inducible shRNA mouse model targeting Atg5 (Atg5i)[13] that phenocopies tissue-specific Atg5 knockout (KO) mice and enables dynamic control of autophagy (Supplementary Figs. 1 and 2). These mice lack brain expression of the shRNA and as such do not suffer from the lethal neurotoxic effects that characterise systemic autophagy knockout mice[14,15], and enable us to perform longitudinal studies that were previously unachievable in vivo.

A common caveat of many mouse models is that genetic manipulations are often present during embryogenesis. Thus, any phenotypes that manifest are a combination of both developmental and tissue homoeostasis effects. To avoid the generation of these compound effects, Atg5i mice were aged until 8-weeks (young adults) before being transferred to a dox-containing diet and followed to assess overall survival. Atg5i mice on long-term dox (LT-Atg5i) had a median survival of ~6 months on dox (Male 185 days; Female 207 days on dox) with no apparent sex bias (Fig. 1a–c and Supplementary Fig. 3a).

In comparison with littermate controls, LT-Atg5i mice experienced a progressive deterioration, initially presenting with a reduction in coat condition within the first few weeks and a reduction in weight gain that became more pronounced over the life of the animal (Fig. 1d, e and Supplementary Fig. 3b). The majority of mice eventually succumbed to a general morbidity characterised by lethargy, piloerection, and a decrease in body condition, wherein they have to be sacrificed. As previously described with naturally aged colonies[16], LT-Atg5i mice also appeared susceptible to eye infections and ulcerative dermatitis, the latter being primarily localised to the ears and neck and ranging from mild to severe (Fig. 1f and Supplementary Fig. 3c, respectively).

A singular cause of death in LT-Atg5i mice is difficult to determine and it is most likely of multifactorial aetiology across the cohort. At necropsy, all mice displayed hepatomegaly and splenomegaly in comparison to age and sex matched controls, consistent with phenotypes associated with tissue specific knock-out mice[17–19] (Supplementary Fig. 3d, e). Elevated serum Alanine Aminotransferase (ALT) and reduced levels of serum albumin were present throughout dox administration of Atg5i mice, yet were altered further at the time of death only in a subset of samples (Supplementary Fig. 3f, g, yellow circles). Consistent with this, an increase in serum bilirubin levels was only observed at the time of death within this same subset of mice (Supplementary Fig. 3h, yellow circle). These data suggest that severe liver failure occurs in only a fraction.

Interestingly serum creatinine levels, a marker of kidney function, also displayed an increase only in a different subset of LT-Atg5i mice at the time of death, although they were not generally elevated during dox administration (Supplementary Fig. 4a). Loss of autophagy also correlated with a general thickening of the basement membrane and the presence of sclerotic (Supplementary Fig. 4b) and enlarged glomeruli (Supplementary Fig. 4c, d) in comparison with age-matched tissue samples, indicative of degenerative kidney disorder. These data suggest that, similar to the liver, systemic autophagy defect causes age-associated degenerative alterations in kidney, yet only a distinct subset progresses to renal failure on death. In addition to this stochastic development of organ failure, LT-Atg5i mice universally presented with cardiomyopathy (Supplementary Fig. 4e). Histological examination highlighted the presence of enlarged, degenerate and vacuolated cardiomyocytes, in addition to the presence of cardiac fibrosis (Fig. 1g).

Together, our data suggest that, despite the stereotypic premature death, LT-Atg5i mice suffered from a heterogeneous set of tissue degenerative disorders that appear to have contributed to an increase in mortality. Of note, there was no evidence of overt tumour development in these mice at the time of death.

**Autophagy inhibition is associated with accelerated ageing.** After 4 months of dox treatment, all LT-Atg5i mice displayed evidence of kyphosis that became progressively more pronounced as the animals aged until death, whilst 16/28 LT-Atg5i mice

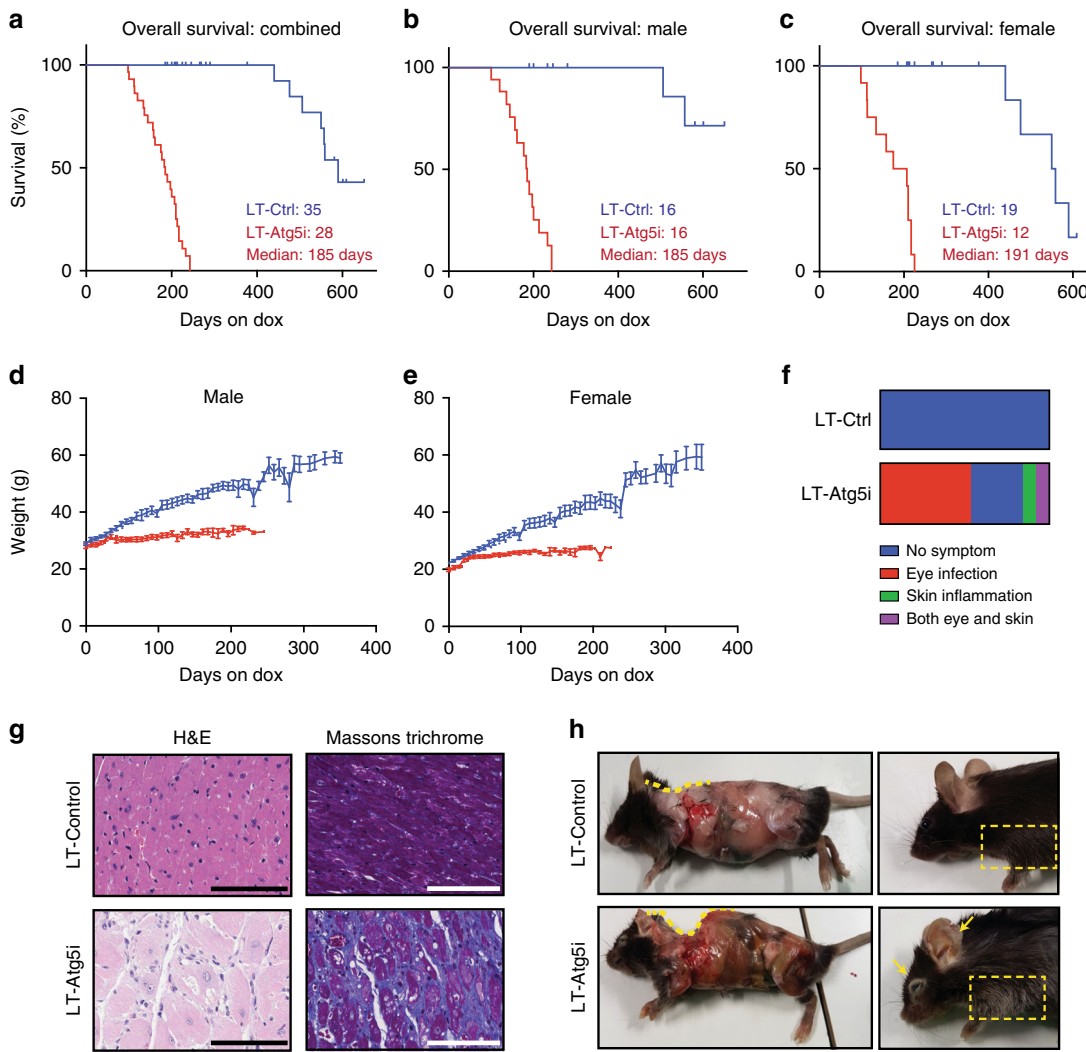

**Fig. 1 Autophagy inhibition decreases lifespan.** LT-Atg5i mice on dox continuously from 2 months old display a reduced lifespan in comparison with LT-Control as shown in survival graphs for combined ($p < 0.0001$) (**a**), male ($p < 0.0001$) (**b**), female ($p < 0.0001$) (**c**) (Mantel–Cox test). Median survival (days on dox) and mice per group are indicated. During this period LT-Atg5i mice also display a reduced weight gain in both male (**d**) and female (**e**) cohorts. **f** LT-Atg5i mice also display an increased frequency of skin inflammation and eye infections in comparison with age-matched LT-Control mice. **g** Cardiac fibrosis was also evident in LT-Atg5i mice. Representative images of H&E and Massons Trichrome are shown. Scale bars, 100 µm. **h** Age-matched skinned mice. LT-Atg5i mice show kyphosis (yellow dotted line traces the arch of the spine). They often displayed premature greying (dotted rectangle). Arrows indicate the presence of inflammation.

displayed evidence of premature greying to varying degrees (Fig. 1h). Furthermore, LT-Atg5i mice displayed evidence of extramedullary hematopoiesis (Fig. 2a) and immune aggregations, commonly seen in aged mouse colonies, were also found in the liver, lungs and kidneys but were generally absent in age matched controls, although the incidence of these increased in frequency with increasing age (Supplementary Fig. 5a–c).

As previously described in hematopoietic Atg5 KO mice, LT-Atg5i mice also displayed an increase in cellularity of the peripheral immune system[18,20] (Fig. 2b, left) with a myeloid skewing (Fig. 2c) reminiscent of age-associated chronic inflammation. This 'inflamm-ageing' phenotype was further supported by an increase in serum TNF and IL-6 in LT-Atg5i mice in comparison with control (Fig. 2d). In addition, serum isolated from LT-Atg5i mice displayed positivity of antinuclear antibodies in 5/12 cases tested in comparison with 1/6 control mice, with the predominant staining pattern being homogeneous and speckled, implying a systemic autoimmune reaction in a subset of autophagy inhibited mice (Supplementary Fig. 5e).

To determine whether the immune phenotypes were driven by autophagy loss in the immune system or due to systemic autophagy loss, we transplanted bone marrow from untreated Control and Atg5i mice into irradiated wild-type C57BL/6 mice. Subsequent doxycycline treatment for 4 months recapitulated the myeloid skewing in peripheral blood in the mice with Atg5i bone marrow (Fig. 2e) but with an apparent decrease in the immune cellularity (Fig. 2b, right). Furthermore, in those mice, there appeared to be a reduction in the donor-derived component (i.e. Atg5i bone marrow-derived) of the peripheral blood (Supplementary Fig. 5g). Largely consistent with a previous study using *Atg12* mutant mice[18], combined these results suggest that the general WBC expansion is driven by systemic autophagy loss, while the myeloid skewing is immune cell intrinsic.

Skeletal muscle exhibits an age-related decline and autophagy has been reported to be required for the maintenance of Pax7 positive satellite cells (myogenic precursors)[21]. In accordance, LT-Atg5i mice displayed evidence of skeletal muscle degeneration with the presence of smaller fibres, a reduction in the population

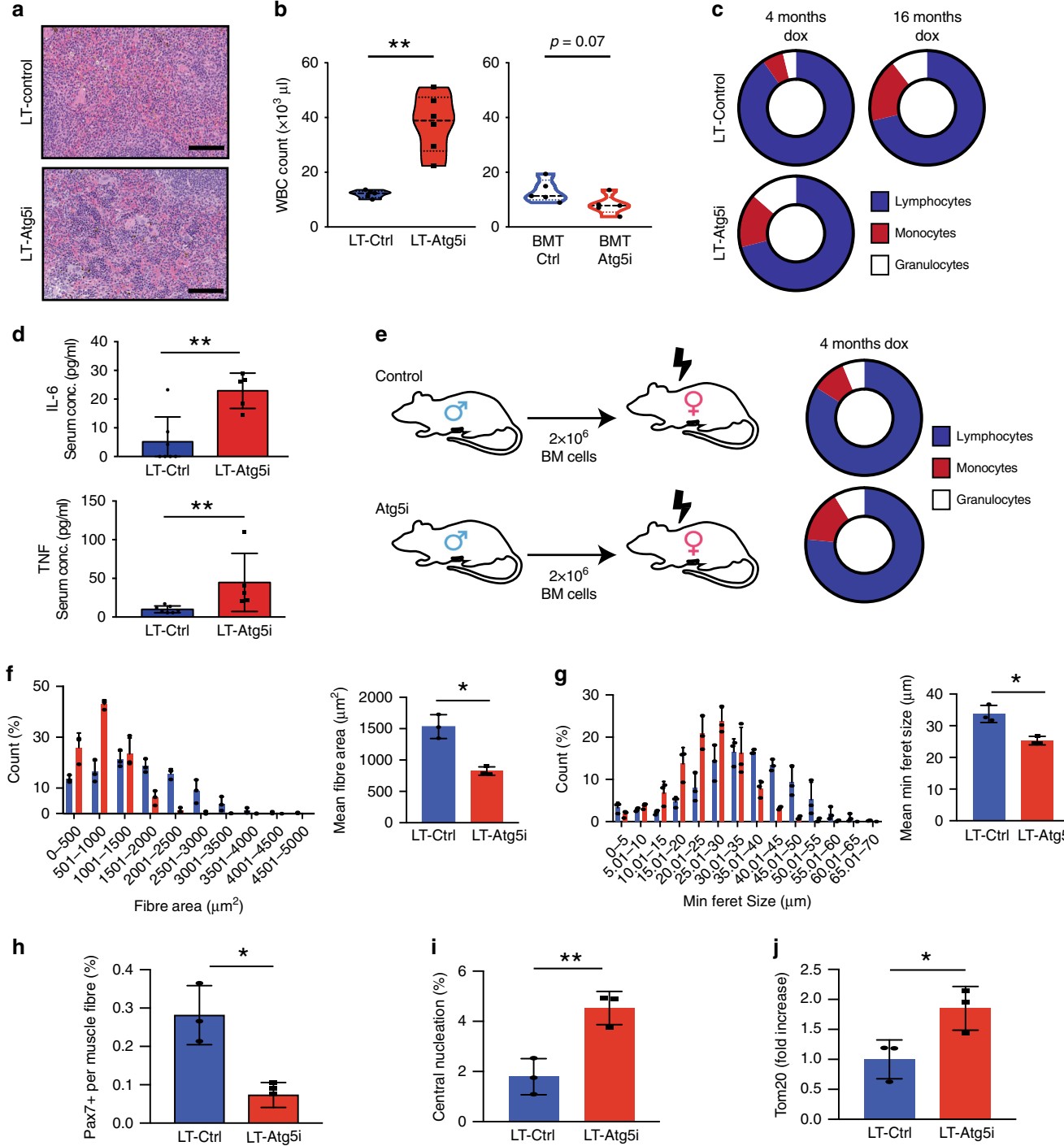

**Fig. 2 LT-Atg5i mice present with accelerated aging phenotypes. a** Extramedullary haematopoiesis is present in the spleens of LT-Atg5i mice in comparison with age-matched controls. Scale bars, 100 μm. **b** 6-month-old LT-Atg5i mice (4 months dox treatment) display increased White Blood Cell counts (WBC). Meanwhile, irradiated wild-type mice in receipt of uninduced bone marrow from Ctrl or Atg5i mice display a reduced WBC count after 4 months of dox treatment (unpaired two-tailed Welches $t$ test, $n = 5$–6 per group). **c** Composition of the peripheral immune system in LT-Atg5i mice is reminiscent of old control mice ($n = 5$–6 mice per group). **d** 6-month-old LT-Atg5i mice (4 months dox treatment) displayed increased serum levels of IL-6 and TNF (LT-Atg5i $n = 5$, LT-Ctrl $n = 7$; Mann Whitney Test). **e** Bone marrow transplantation of uninduced Ctrl and Atg5i bone marrow into irradiated wild-type recipient mice after 4 months of dox treatment Atg5i recipient mice display a myeloid skewing. LT-Atg5i mice display alterations in skeletal muscle after 4-month of dox treatment. LT-Atg5i mice display a significant difference in cross-sectional area (**f**) ($n = 3$ R-Ctrl and 3 R-Atg5i, unpaired two-tailed Welch's $t$ test) and minimum feret size (**g**) ($n = 3$ R-Ctrl and 3 R-Atg5i, unpaired two-tailed Welch's $t$ test). LT-Atg5i mice also display a decrease in Pax7 nuclear positivity per fibre (**h**), an increase in central nucleation (**i**), and positivity for the mitochondrial marker TOM20 (**j**), as determined by tissue immunofluorescence (unpaired two-tailed Welches $t$ test; $n = 3$ R-Ctrl and 3 R-Atg5i). Error bars indicate standard deviations. *$p < 0.05$; **$p < 0.01$, ***$p < 0.001$.

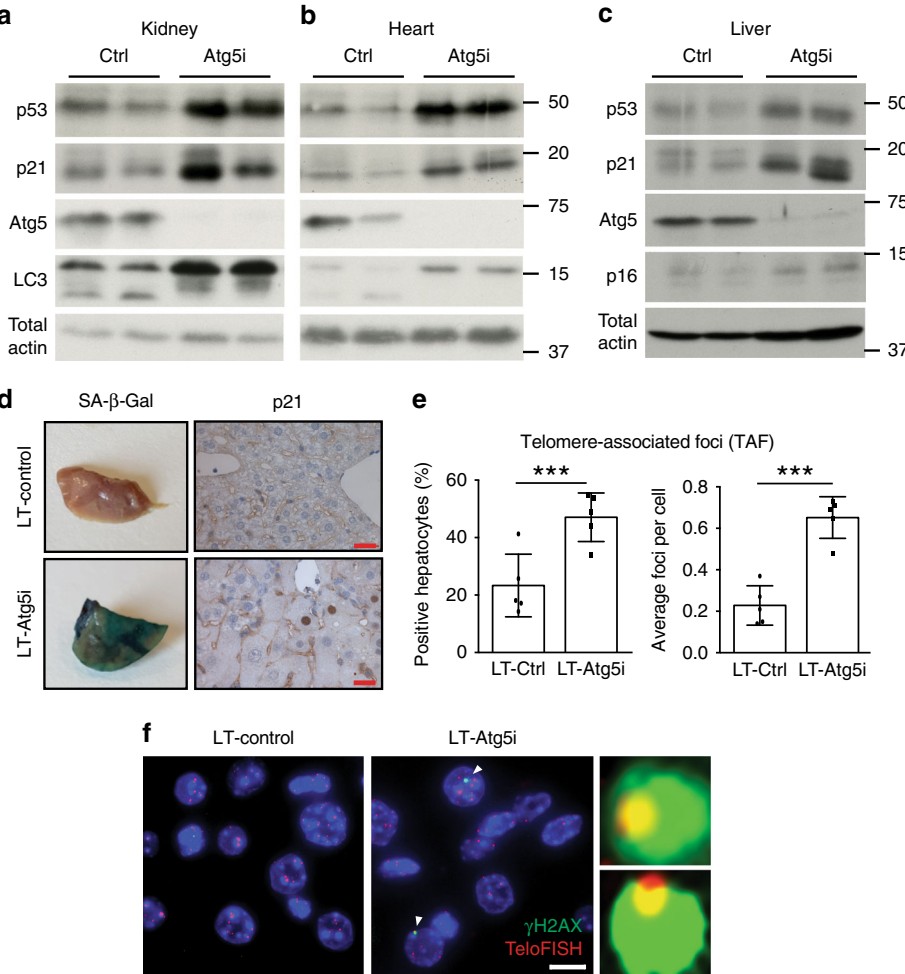

**Fig. 3 Autophagy inhibition drives senescence in vivo.** Markers of senescence can also be seen across multiple tissues in our LT-Atg5i cohorts treated with dox for 4 months including in kidney (**a**), heart (**b**), and liver (**c**). LT-Atg5i livers stain positively for senescence associated β-galactosidase and p21 unlike age-matched control mice (**d**) (scale bar, 25 μm). **e** 6-month doxycycline treated LT-Atg5i livers display an increase in the frequency and abundance of γ-H2AX at telomeres, a marker associated with increasing chronological age (unpaired two-tailed *t* test; *n* = 5). **f** A representative example image shown. Arrowheads point to TAF that are magnified on the right of the image. Scale bar, 10 μm. Error bars indicate standard deviation ***p < 0.001. For **a–c** source data are provided as a Source Data file.

of Pax7 positive satellite cells, and an increase in central nucleation in comparison with age-matched littermate control mice (Fig. 2f–i, Supplementary Fig. 6a, b). Central nucleation represents muscle fibre regeneration after acute muscle injury but an increase in basal frequency of centrally nucleated myofibres is also a sign of sarcopenia at geriatric age both in mice and human[22]. In addition, LT-Atg5i muscle fibres displayed increased staining positivity for the mitochondrial marker Tom20 indicative of increased mitochondrial mass and a reduction in autophagy mediated turnover (Fig. 2j).

The accumulation of senescent cells is considered a key marker of chronological ageing. Autophagy has been reported to have context dependent and sometimes opposing roles during cellular senescence: typically basal autophagy is considered to promote fitness and its loss may promote senescence, whereas in oncogene-induced senescence, autophagy may be important for the establishment of senescent phenotypes[23–26]. To determine if the systemic loss of basal autophagy is sufficient to drive the establishment of cellular senescence in vivo, we performed western blotting across a number of tissues from 4-month dox treated LT-Atg5i mice and found an increased staining pattern for key senescence markers (i.e. p16, p21, and p53) (Fig. 3a–c and Supplementary Fig. 6c). In addition, whole mount senescence-

associated beta-galactosidase staining from 6-month treated livers highlighted a marked increase in staining patterns in comparison with LT-Control mice (Fig. 3d). Histologically, nuclear accumulation of p21 was also evident, particularly in hepatocytes with enlarged morphology (Fig. 3d). Furthermore LT-Atg5i mice display a significant increase in both the abundance and frequency of telomere-associated γ-H2AX foci (TAF) in liver, lung and heart tissue (Fig. 3e, f and Supplementary Fig. 6d, e). TAF represent persistent damage in telomeric regions, independent of length, that are resistant to repair machinery and have been shown to correlate with senescence, increasing age and mitochondrial dysfunction[27–29]. The increase in TAF abundance therefore reinforces the notion that mice exhibit age acceleration upon systemic autophagy reduction.

Of note, similar gross phenotypic results were also seen in mice with a second hairpin targeting Atg5 (LT-Atg5i_2). LT-Atg5i_2 mice display evidence of premature ageing-like phenotypes (Supplementary Fig. 7a–c), however the appearance of these phenotypes was delayed in comparison with LT-Atg5i mice, seemingly due to a hypomorphic phenotype. Accordingly, these mice displayed the accumulation of p62/Sqstm1 and LC3 in multiple tissues but at lower levels in comparison with LT-Atg5i mice, and did not display phenotypes associated with complete

Atg5 knockout mice, including hepatomegaly and splenomegaly (Supplementary Fig. 7d–f). These findings in particular are important as they establish that the reduction in longevity and presence of ageing phenotypes is not dependent on the hepatomegaly and splenomegaly phenotypes encountered in the original LT-Atg5i mouse strain with the highest degree of autophagy inhibition.

Combined these data support a role for basal autophagy in maintaining tissue and organismal homoeostasis and provide evidence that causally links autophagy inhibition to the induction of ageing-like phenotypes in mammals.

**Autophagy restoration partially reverses ageing phenotypes.** We next sought to determine whether autophagy restoration alone is able to reverse the ageing-like phenotypes by removing dox from the diet. 8-week old Atg5i and control mice treated with dox for 4 months, the point at which they universally presented with kyphosis, were switched back to a diet absent of dox leading to a restoration in Atg5 levels and autophagy (termed R-Atg5i cohort) (Fig. 4a, b and Supplementary Fig. 8a)[13]. Interestingly, while p16 levels reduced in the livers R-Atg5i mice, they still appeared elevated in comparison with age-matched control mice 4-months post dox removal (Fig. 4b). This is in contrast to the kidney that exhibited only a mild increase in p16 that was mostly reversed upon autophagy restoration. While further systematic analyses would be required, the data suggest a differential susceptibility to autophagy inhibition across organs.

An increase in chronological age is generally associated with the deviations in multiple health parameters that when measured can be combined into a clinical 'frailty-score'[30]. As expected, R-Atg5i mice displayed an initial increase in their frailty scores during autophagy inhibition in comparison with littermate controls, yet once mice have been switched back to a diet absent of dox, the frailty scores displayed a significant decrease over the next 4 months (Fig. 4c, Supplementary Movie 1). In contrast, LT-Atg5i mice treated on dox for 6 months (median survival is around ~6 months on dox) continued to display a significant difference in their frailty scores, while almost all LT-Atg5i mice had already succumbed by 8-months (Fig. 4c). A similar increase in frailty was also noted in the LT-Atg5i_2 cohorts (Supplementary Fig. 7b). The penetrant kyphosis phenotype was largely irreversible, however 3/26 R-Atg5i mice did show evidence of recovery from kyphosis, while no mice displayed a reversal of the greying phenotypes. As such, while autophagy inhibition in vivo appears to promote frailty, autophagy restoration is seemingly able to substantially reverse this effect.

Remarkably the profound immune-associated phenotypes that we observed in autophagy-deficient LT-Atg5i mice were reversed in R-Atg5i mice. Serum markers of inflammation and white blood cell counts were indistinguishable between R-Atg5i and R-Control mice (Fig. 4d, e and Supplementary Fig. 8b). However, it should be noted that, in aged R-Atg5i mice removed from dox for 8 months (14 months old), there was a trend towards a larger red blood cell distribution width (RDW), which has previously been linked to a range of diseases and an increased risk of acute myeloid leukaemia (AML) (Fig. 4f)[31]. In addition, R-Atg5i livers displayed a complete reversal of hepatomegaly and serum ALT levels (Supplementary Fig. 8c, d). The kidneys of R-Atg5i mice appeared to recover from autophagy inhibition and lacked evidence of sclerotic and enlarged glomeruli (Supplementary Fig. 8e–g). Consistently, serum albumin levels displayed evidence of normalisation, although there was still a trend for reduced levels in R-Atg5i mice at the time point tested, suggesting that liver and/or kidney functions are largely recovered, if not completely (Supplementary Fig. 8h).

Similarly, the protein aggregation marker p62/SQSTM1 in the liver appeared much reduced in R-Atg5i mice in comparison to the LT-Atg5i mice, yet a small but substantial number of cells still exhibited a marked accumulation of p62 aggregation in R-Atg5i mice that had been off dox for 4 months (Fig. 5a). In addition, R-Atg5i livers were also found to contain the presence of ceroid-laden macrophages and lipofuscin positivity, pigments known to increase with age and not seen in age-matched controls mice (Fig. 5b). Importantly, and in accordance with this partial restoration phenotype, molecular markers of ageing such as TAF also remained significantly elevated in R-Atg5i mice (Fig. 5c). This is consistent with the persistent nature of telomeric DNA damage, which is reported to be irreparable[27,32]. Together with other senescence markers (Fig. 4b), these data suggest that a portion of the cellular damage caused by a chronic block in autophagy is irreversible.

Analysis of skeletal muscle from R-Atg5i mice, with autophagy restoration, suggests that muscle fibre size, morphology, and satellite cell frequency display no sign of recovery 2 months post dox removal (Fig. 5d–f and Supplementary Fig. 6b). However, central nucleation frequency was dramatically reduced and comparable with control (Fig. 5g). As expected with Atg5 restoration, Tom20 positivity appeared similar to control levels (Fig. 5h). In addition, the cardiac fibrosis observed LT-Atg5i mice appears to still be present 4 months post dox removal in R-Atg5i cohorts (Supplementary Fig. 9c). Together these data suggest that autophagy restoration may have tissue and pathology specific limitations in the capacity to recover from the tissue and cellular damage induced upon its inhibition. Crucially, whilst some tissues, such as the liver, appear to recover, they are still exhibit age-associated pathologies at the molecular level.

**Accelerated tumour development in R-Atg5i mice.** As R-Atg5i mice displayed some evidence of organismal rejuvenation and an increase in overall health, we sought to determine if autophagy restoration is able to reinstate natural longevity to the level seen in littermate control mice, or whether the damage accumulation impacting on lifespan was irreversible. Remarkably, the life-span of R-Atg5i mice was significantly extended in comparison with LT-Atg5i mice (median survival 493 days versus 185 days since treatment began, respectively), while it was still significantly shorter than the R-Control cohorts (Fig. 6a). In marked contrast to LT-Atg5i mice, the cause of death was predominantly associated with the development of tumours with an increased frequency and at earlier timepoints (Fig. 6b, c). These tumours display no evidence of continued autophagy inhibition via immunohistochemical (IHC) analysis (Fig. 6d). Of note a whole-body mosaic Atg5 knockout mouse model has been previously reported to only develop liver adenomas but without any malignant tumours[33]. Together, our data suggest that a temporary period of autophagy inhibition may be enough to induce irreversible cellular damage, which might facilitate tumour development cooperatively with the restoration of autophagy.

## Discussion

While the rate of autophagic flux is believed to decrease with advancing age and has been postulated to be a driver of ageing in multicellular organisms, evidence in mammals has been limited to the role of autophagy in maintaining stem cell populations[18,21]. Such systemic organismal studies have been impossible to conduct owing to the embryonic or neonatal lethality and, in adult mice, rapid neurotoxicity, which accompany systemic autophagy ablation[14,34]. The temporal control and lack of brain shRNA expression afforded by the Atg5i model have enabled us to circumvent these barriers, and separate developmental from tissue

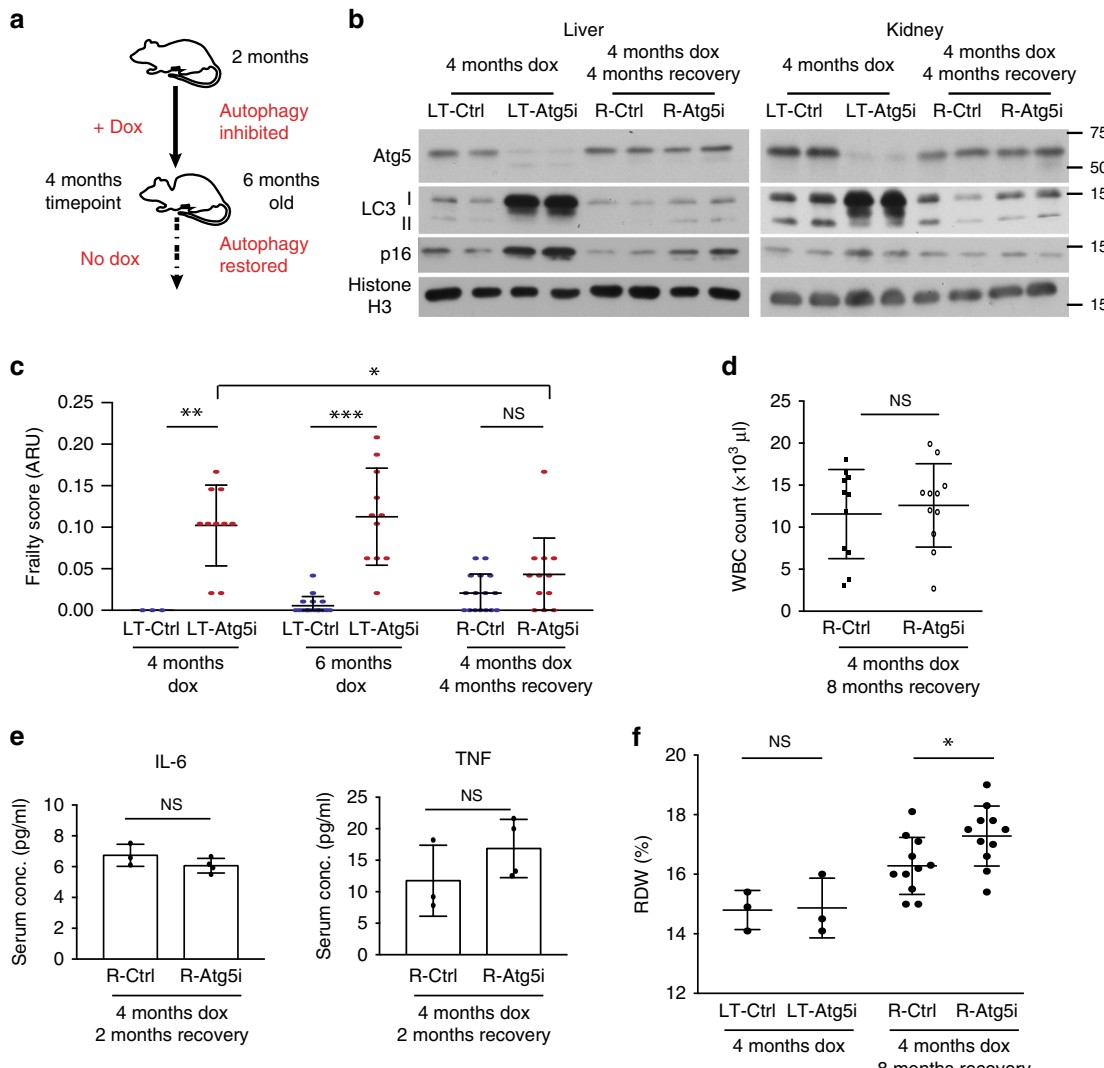

**Fig. 4 Restoration of autophagy partially restores health-span. a** Schematic of R-Atg5i study. Briefly 2-month old mice are given dox to induce Atg5 downregulation for 4 months at which point they exhibit ageing-like phenotypes. Dox is then removed and autophagy restored. **b** Liver and kidney tissues from R-Atg5i mice with autophagy restored for 4 months display evidence of Atg5 protein and autophagic flux restoration, yet the liver still stains positively for the marker of senescence p16. **c** Atg5i mice on dox for 4 months and 6 months display increase frailty scores in comparison with controls (ARU, arbitrary units). While R-Atg5i mice where autophagy has been restored for 4 months, display a recovery (Two-way ANOVA with Tukey's correction for all comparisons, $n = 3$–16). **d** Whole blood cell counts from R-Atg5i mice display no difference in comparison with age matched R-Control mice (unpaired two-tailed $t$ test; $n = 11$ per group). **e** Inflammatory serum cytokines IL-6 and TNF are equivalent in R-Atg5i and R-Control mice 2-months post dox removal (Mann Whitney test; $n = 3$ R-Ctrl and 4 R-Atg5i). **f** Red blood cell distribution width (RDW) is unaltered in LT-Ctrl and LT-Atg5i cohorts (unpaired two-tailed $t$ test; $n = 3$ per group), yet appears increased in autophagy-restored cohorts in comparison with age-matched littermate control mice (4 months dox, 8 months restoration) (unpaired two-tailed $t$ test; $n = 14$ per group). Error bars indicate standard deviation; NS denotes not significant. *$p < 0.05$; **$p < 0.01$, ***$p < 0.001$. For **b** source data are provided as a Source Data file.

homoeostatic effects that cannot be distinguished in ageing models based on constitutive or in utero genetic modifications.

In addition, it should be noted that whilst the LT-Atg5i model leads to a dramatic reduction in Atg5 levels, with phenotypic consequences of autophagy inhibition being evident (including splenomegaly, hepatomegaly, LC3-I and p62 build-up), they certainly retain some levels of autophagic flux, distinguishing them from the Atg5 KO models. Of note the second hairpin mouse model, LT-Atg5i_2 displays a reduction in Atg5 levels but with a reduced build-up of LC3 and p62, as determined by IHC, and no evidence of hepatomegaly and liver dysfunction, suggesting that this model is hypomorphic. Hypomorphic models may more closely recapitulate the aetiology of human disease, wherein insufficient autophagic flux, not complete block is

associated with pathogenesis and ageing. In addition, the establishment of premature ageing phenotypes in the LT-Atg5i_2 model, without the overt tissue damage (e.g. hepatomegaly), reinforces that reduced autophagy activity, not the liver damage, is the primary driver. However, the widespread perturbation of autophagy across multiple tissues, and the associated dysfunction that accompanies it, almost certainly contributes to the accelerated ageing phenotypes.

Our findings support the theory that a reduction in autophagy is sufficient to induce several molecular and phenotypic characteristics associated with mammalian ageing, including the development of age-associated diseases and a reduction in longevity. Here it is notable that our Atg5i mice phenocopy other models of ageing driven by the accumulation of damage and in

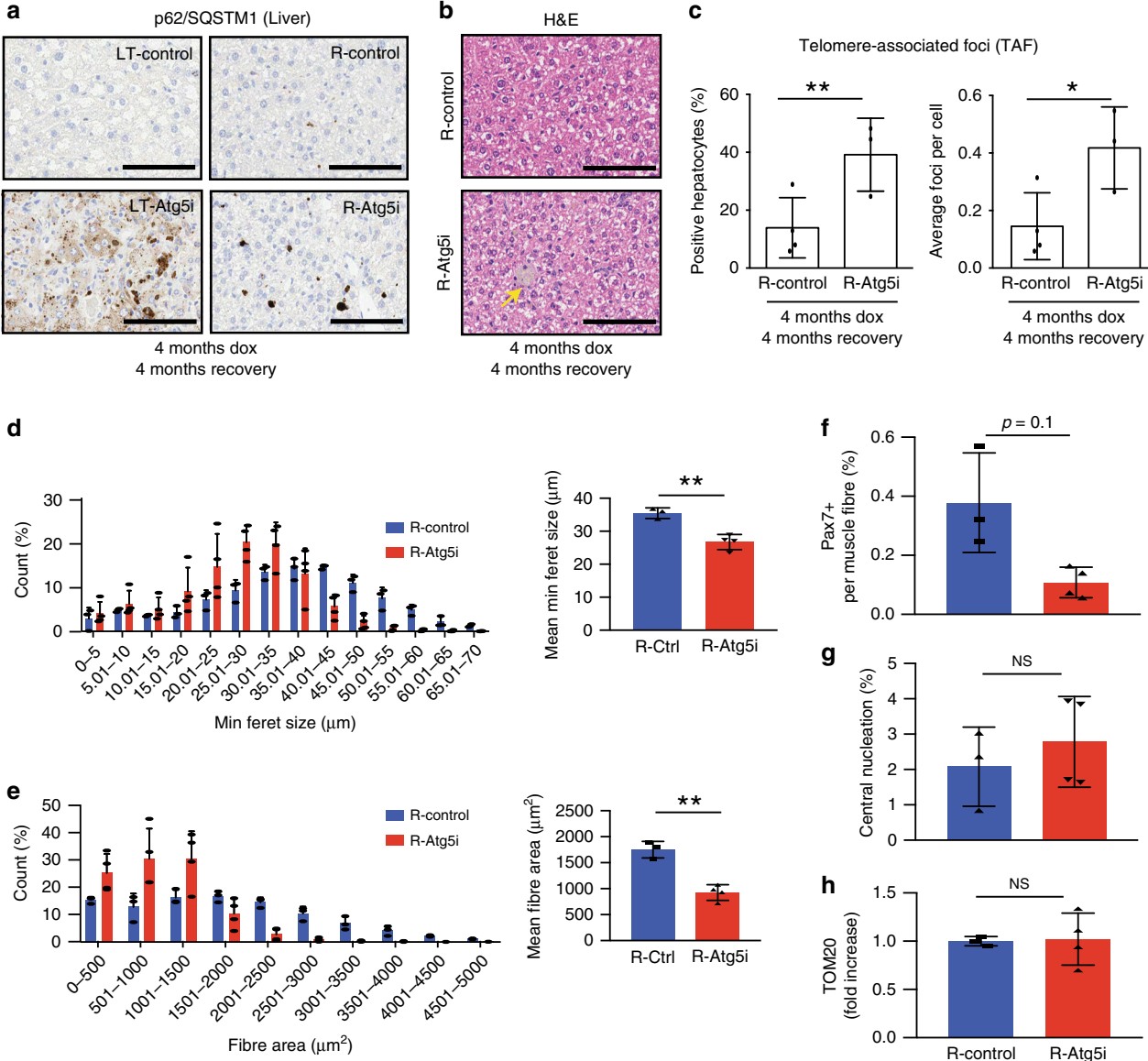

**Fig. 5 Restoration of autophagy does not reverse all markers of aging. a** p62/Sqstm1 staining of R-Atg5i liver highlights the incomplete removal of aggregates 4 months after autophagy restoration. Scale bars, 100 μm. **b** The same livers have a higher incidence of age associated pigmentation in comparison with age-matched control mice. (yellow arrow). **c** TAF frequency and abundance also remains elevated in R-Atg5i mice (unpaired two-tailed *t* test; *n* = 4 R-Ctrl and 3 R-Atg5i). Skeletal muscle analysis from 4 months dox treated and 2 months restored R-Atg5i mice. R-Atg5i muscle fibres continue to display significant alterations in minimum feret size (**d**) (*n* = 3 R-Ctrl and 4 R-Atg5i, Mann Whitney test) and cross-sectional area (**e**) (*n* = 3 R-Ctrl and 4 R-Atg5i, Mann Whitney test). Whilst Pax7 nuclear positivity per fibre still displays no evidence of recovery (**f**), both central nucleation (**g**) and positivity for the mitochondrial marker Tom20 (**h**) exhibit levels similar to R-Ctrl mice. (**f**–**h**, unpaired two-tailed Welches *t* test; *n* = 3 R-Ctrl and 4 R-Atg5i). Error bars indicate standard deviations. *\*p < 0.05; \*\*p < 0.01; \*\*\*p < 0.001.*

particular mitochondrial dysfunction[35,36], however it remains to be seen whether mitochondrial function is altered in this setting. In addition, we cannot rule out synergistic effects of doxycycline side-effects with autophagy inhibition, as such comparison with other inducible models would be required to exclude this possibility.

Several health and life-span extending regimens in mammals, such as calorie restriction or pharmacological modulation, have been posited to exert their effects through the regulation of autophagy[7,37]. However, these effects are also pleiotropic in nature and alter a multitude of cellular processes, making it impossible to deconvolute and ascribe the role of autophagy in these settings. Whilst recent genetic models that promote autophagic flux continuously throughout life have demonstrated

an extension of health- and life-span in mammalian systems[11,12], it is unclear if the damage established by a loss of autophagy is sufficient for age acceleration and can be reversed. If therapeutic regimens in humans are to be established later in life, once autophagy-associated damage has accumulated, ascertaining the capacity for autophagy restoration to repair this damage is critical. In our model, systemic inflammation and frailty scores displayed a marked improvement upon autophagy restoration, which resulted in increased survival. However, while some tissues (i.e. liver and heart) displayed macroscopic normalisation, further analysis highlighted the persistence of pathological phenotypes. Our results indicate that markers of ageing such as TAF, or macroscopic phenotypes such as greying and kyphosis may not fully recover. It should also be noted that we have chosen a late

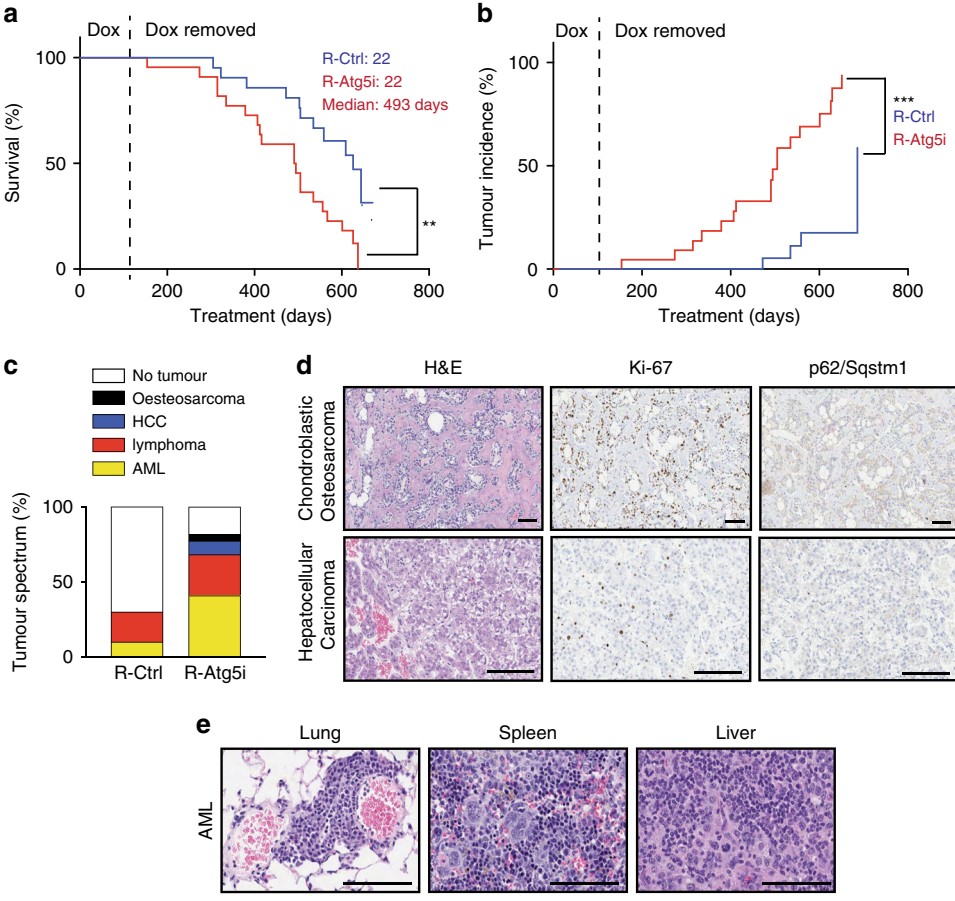

**Fig. 6 R-Atg5i mice are associated with accelerated spontaneous tumour development. a** R-Atg5i mice on display a reduced lifespan in comparison to R-Control mice ($p < 0.01$). **b** Increased frequency of spontaneous tumour formation in R-Atg5i cohorts ($p < 0.001$). **c** Tumour spectrum in R-Atg5i mice versus R-Control mice. **d**, **e** Examples of R-Atg5i tumour histology. H&E staining and immunostaining of indicated proteins. Scale bars, 100 μm.

time-point to restore autophagy as this provided a clear and ubiquitous distinction between control and autophagy inhibited mice, shorter time points or intermittent dosing regimens may display further heterogeneity in damage and recovery phenotypes.

Our unexpected finding, that the temporal inhibition of autophagy predisposes to increased tumour development, provides a potential genetic explanation for the context-dependent role of autophagy in tumorigenesis:[38,39] i.e. autophagy can be a tumour suppressor[33,40,41] or a tumour promoter[42–44]. The irreversible damage induced by autophagy inhibition (e.g. genomic instability), might confer tumour susceptibility, while autophagy activity is perhaps required for actual malignant transformation. The clinical implication of our data is not limited to the advanced age state. As some pathophysiological states, such as obesity, are associated with an insufficient level of autophagy[45], it would be interesting to determine if obese individuals retain an increased risk of tumour development even upon weight loss, in comparison with never obese populations.

## Methods

**Atg5i mouse maintenance and aging**. The generation and initial characterisation of the Atg5i transgenic line have previously been described in detail[13]. Mice were maintained on a mixed C57Bl/6 × 129 background with littermate controls used in all experiments. All experimental mice were maintained as heterozygous for both the shRNA allele and CAG-rtTA3 alleles, whereas control littermates were lacking one of the alleles. Guide sequences were as follows: Atg5i (Atg5_1065) TATGAAG AAAGTTATCTGGGTA[13]; Atg5i_2 (Atg5i_1654) TTATTTAAAAATCTCTCACT GT. Atg5_1654 was chosen after an initial screen for shRNA knockdown efficiency wherein it displayed the second highest efficiency of knockdown[13]. The shRNA guides in a miR-E design were inserted downstream of the *Col1a1* locus via recombinase-mediate cassette exchange which enables efficient targeting of a

transgene to a specific genomic site 500 base pairs downstream of the 3′UTR in D34 ES cells. Mice were maintained in a specific pathogen-free environment under a 12-h light/dark cycle, having free access to food and water. These mice were fed either a laboratory diet (PicoLab Mouse Diet 20, 5R58) or the same diet containing doxycycline at 200 ppm (PicoLab Mouse Diet, 5A5X). For this study mice were aged for 2 months before doxycycline administration in the diet. Mice were enroled either to time-point study groups or long-term longevity cohorts (LT- and R-groups). Experienced animal technicians checked mice daily in a blinded fashion, and additionally mice were weighed and hand-checked on a weekly basis. Mice found to be of deteriorating health were culled under the advice of senior animal technicians if displaying end of life criteria. These signs include a combination of (1) hunched body position with matted fur, (2) piloerection, (3) poor body condition (BC) score (BC1 to 2), (4) failure to eat or drink, (5) cold to touch, and or (6) reduced mobility, including severe balance disturbances and ataxia. In accordance with UK home office regulations any mice suffering a 15% loss of body weight were also considered to be at an end-point. Note that for LT- longevity cohorts a portion of control mice were culled to generate age-matched littermate control tissue. These mice are marked as censored events on the survival curve. For analysis, mice were treated as alive up to the point of their removal from the study where they are considered lost to follow-up and are not included in the calculations of median longevity. All experiments were performed in accordance with national and institutional guidelines, and the ethics review committee of the University of Cambridge approved this study.

**Frailty scoring**. Clinical frailty scoring was determined using the previously published frailty index[30]. A blinded researcher and animal technician performed all frailty scores independently within the same 48 h period and scores were compared afterwards to ensure accuracy of phenotype scoring. The method is based on scoring 31-parameters as Normal (scores 0), Mild (scores 0.5), or Severe (scores 1). The total score for a mouse is then divided by the number of metrics being analysed to create a total frailty score for the animal. This includes alopecia, loss of fur colour, dermatitis, loss of whiskers, coat condition, presence of tumours, distended abdomen, kyphosis, tail stiffening, gait disorders, tremors, forelimb grip strength, body condition score, vestibular disturbance, hearing loss, presence of cataracts, alterations to corneal opacity, eye swelling or discharge, sunken eyes (one

or both), vision loss, menace reflex, nasal discharge, malocclusions, rectal prolapse, prolapse (vaginal, uterine, or penile), diarrhoea, altered respiratory rate, alterations to mouse grimace, piloerection, body temperature and weight.

**Doxycycline serum measurements**. An LC-MS/MS assay was developed for the analysis of doxycycline in mouse plasma with demeclocycline as an internal standard.

Doxycycline Hyclate (Sigma-Aldrich, 108M4031V) and Demeclocycline HCl (Sigma-Aldrich, D61140) were purchased (Sigma-Aldrich) and individual stock solutions were prepared in water: methanol: formic acid (9:1:0.1) to a concentration of 1 mg/mL of the free base. Doxycycline calibration standards were prepared in K2 EDTA mouse plasma, with a final range of 0.5–125 ng/mL. Ten microliter of sample was mixed with 10 μL of internal standard (25 ng/mL in water:methanol: formic acid (9:1:0.1)) and extracted with 100 μL of ethyl acetate. The organic layer was transferred, evaporated and reconstituted in 50 μL of water:methanol:formic acid (9:1:0.1). 5 μL was injected into the LC-MS/MS system. Chromatography was performed on a Shimadzu Nexera X2 UHPLC system with a Phenomenex Luna Omega C18 100 Å 1.6 μm 100 × 2.1 mm column at 35 °C using a 0.1% formic acid in water/0.1% formic acid in acetonitrile gradient at 0.4 mL/min over 5 min. Doxycycline and demeclocycline had retention times of 1.87 and 1.81 min respectively. Doxycycline produced an unavoidable split peak, but it was reproducible, consistent and did not affect the precision or accuracy. The liquid chromatograph was coupled to a Sciex Triple TOF 6600 mass spectrometer operated using positive electrospray ionisation and enhanced mass high sensitivity product ion scan mode for doxycycline ($m/z$ 445.2–28.1360) and demeclocycline ($m/z$ 465.07–448.0810). Data acquisition was controlled via Sciex Analyst TF 1.7.1 software and data processed using Sciex MultiQuant 3.0.2 with a processing mass peak width of 0.05 Da for both doxycycline and demeclocycline (i.e. 428.1360 Da ± 0.025 and 448.0810 Da ± 0.025 respectively). A linear $1/x^2$ weighted regression using the peak area ratio of doxycycline and demeclocycline was used to construct the calibration curve. Precision and accuracy were within the predefined criteria of ±20%.

**Pathology and Immunohistochemistry**. Explanted tissues were fixed in 10% neutral-buffered formalin solution for 24 h and transferred to 70% ethanol. Tissues were embedded in paraffin, cut in 3 μm sections on poly-lysine coated slides, deparaffinized, rehydrated, and stained with H&E. The PAS, Congo Red and Massons Trichrome histochemical stains were performed according to established protocols. An experienced pathologist reviewed all histology blinded for evidence of tumours and tissue pathologies. For immunohistochemistry and tissue immunoflourescence formalin-fixed paraffin-embedded samples were de-waxed and rehydrated. For anti-P21 (Santa Cruz, SC-6246; 1:500), and anti-TOM20 (Santa Cruz SC-11415, 1:500) staining antigen unmasking was performed with citrate buffer (10 mM sodium citrate, 0.05% Tween 20, pH 6) in a pressure cooker for 5 min at 120 °C. For P21 exogenous peroxidases were quenched in 3% $H_2O_2$/PBS for 15 min and the remaining steps were performed according to Vector Labs Mouse on Mouse staining kit (MP-2400). The remaining antibodies were used at the following concentrations and ran on the Leica Polymer Detection system (DS9800) with the Leica automated Bond platform: Anti-SQSTM1 (Enzo, BML-PW9860; 1:750), anti-KI67 (Bethyl Laboratories, IHC-00375; 1:1000), Anti-LC3 (Nanotools, LC3-5F10 0231-100, 1:400). Anti-CD45-B220 (R&D Systems, MAB1217, 0.67 μg/mL), Anti-CD3 (Dako, A0452, 1:1000), Anti-F4/80 (Serotec, MCA497, 1:20).

For CD45-B220, CD3, F4/80 quantification whole tissue sections were analysed using ImageScope™ (Leica Biosystems). For CD45-B220 and CD3 the percentage positive nuclei were determined. For F4/80 a percentage-positive pixel count was quantified.

For Tom20 analysis the intensity of signal per entire muscle section was determined and an average measurement of intensity per unit area calculated. Samples were then plotted as a fold increase relative to the average intensity per unit of control muscle sections

For kidney glomeruli size tissue sections were analysed using ImageScope™ (Leica Biosystems) and the cross-sectional area of ten glomeruli in the renal cortex was reported per sample.

**Electron microscopy**. Briefly, each mouse was perfused using a Peristaltic Pump P-1 (GE Healthcare) with 50 mL of wash buffer (10 mM PIPES pH 7.4, 137 mM NaCl, 2.7 mM KCl, 2.5 mM CaCl₂, 19.4 mM glucose, 10 mM sodium nitrite, 0.075 mM PVP10), followed by 100 mL of fixative (2% glutaraldehyde/2% Formaldehyde in 0.1 M PIPES pH 7.4 (+2 mM CaCl and 0.075 mM PVP10)). After perfusion tissue was dissected and cut into 1 mm³ before being placed in fixative overnight. These were washed in 0.05 M Na cacodylate buffer (5×), before osmication for 3 days at 4 °C (1% OsO4, 1.5% potassium ferricyanide, 0.05 M Na cacodylate buffer pH 7.4). This was followed with 5× washes in deionized water, a second round of osmication (1 h at room temperature; 2% OsO4 in DIW), and 5× washes in DIW, before samples were passed through a dehydration gradient (3 × 50%, 3 × 70%, 3 × 95%, 3 × 100% ethanol for 5 min each). Samples were then dehydrated in 2 × 5 min washes of 100% acetone, followed by 3 × 5 min washes of 100% acetonitrile. Samples were next placed in Quetol resin mix (12 g Quetol 651,

15.7 g NSA, 5.7 g MNA, 0.5 g BDMA) added to equal volumes of 100% acetonitrile for 24 hrs at room temperature. After which samples were placed into pure Quetol resin mix (with BDMA) for 5 days, with fresh resin mixed added daily. Embedded samples were placed into moulds and incubated at 60 °C for 48 h before being sectioned (~80 nm) on an ultramicrotome (Leica Ultracut) and mounted onto 400 mesh bare copper grids. TEM was performed on a FEI Tecnai G20 electron microscope run at 200 keV accelerating voltage and using a 20 μm objective aperture to improve contrast.

**Western blotting**. Tissue samples were homogenised with the Precellys 24 tissue homogeniser in Laemmli buffer and samples ran on 12.5 or 15% gels. Protein was transferred to PVDF membranes (Immobilon, Millipore), which was subsequently blocked for 1 h at room temperature (5% milk solution in TBS-Tween 0.1%) before incubating with primary antibody at 4 °C overnight. An appropriate HRP-conjugated secondary antibody was incubated at room temperature for 1 h. Western blots were visualised with chemiluminescence reagents (Sigma, RPN2106). Antibodies were used at the following concentrations: Anti-ATG5 (Abcam, ab108327; 1:1000), anti-LC3 (Abcam, ab192890; 1:1000), anti-ACTIN (Santa Cruz Biotechnology, I-19; 1:5000 [no longer commercially available]), anti-P53 (Cell Signalling Technologies, Clone 1C12; 1:1000), anti-P21 (Santa Cruz, SC-6246; 1:1000), anti-Histone H3 (Abcam, ab1791; 1:5000), anti-P16 (Santa Cruz, SC-1207; 1:1000), anti-HMGA1 (Abcam, ab129153; 1:1000), anti-NBR1 (Abcam, ab55474; 1:1000).

**Blood and serum analysis**. Whole blood composition was performed using the Mythic Haematology Analyser to determine whole blood counts, immune composition, and RDW. Mouse cytokines were determined using a cytometric bead array (BD Biosciences, Catalogue number: 552364). Sera isolated from mice were analysed by the Core Biochemical Assay Laboratory (CBAL), Cambridge, UK for Alanine Transferase (Siemens Healthcare), Albumin (Siemens Healthcare), Bilirubin (Siemens Healthcare), and Creatinine (Siemens Healthcare) using automated Siemens Dimension RxL and ExL analysers.

**Anti-nuclear antibody detection in HEp-2 cells**. Serum samples from control and Atg5i mice were diluted 1:50, 1:100 and 1:200 with PBS. The diluted sera were incubated with human epithelial cell (HEp-2) substrate slides (Kallestad Bio-Rad #26102) for 30 min at room temperature in a humidified chamber. After 3 × 5 min washes in PBS, samples were blocked with 5% normal goat serum for 1 h and subsequently incubated with AlexaFluor488 conjugated goat anti-mouse IgG antibody in 5% normal goat serum for 1 h. The slides were then washed as previously and were evaluated using fluorescence microscopy. Interpretation of positivity and grading were performed using the ×20 objective while evaluation of pattern was performed using the ×40 objective.

**Telomere associated DNA damage foci (TAF)**. Formalin-fixed paraffin-embedded liver sections were hydrated by incubation in 100% Histoclear, 100, 95 and 2 × 70% methanol for 5 min before washed in distilled water for 2 × 5 min. For antigen retrieval, the slides were placed in 0.1 M citrate buffer and heated until boiling for 10 min. After cooling down to room temperature, the slides were washed 2× with distilled water for 5 min. After blocking in normal goat serum (1:60) in BSA/PBS, anti-γ-H2A.X primary antibody (Cell Signalling Technologies, S139; 1:250) was applied and incubated at 4 °C overnight. Slides were washed 3× in PBS, incubated with secondary antibody for 30 min, washed three times in PBS and incubated with Avidin DCS (1:500) for 20 min. Following incubation, slides were washed three times in PBS and dehydrated with 70, 90 and 100% ethanol for 3 min each. Sections were denatured for 5 min at 80 °C in hybridisation buffer (70% formamide (Sigma), 25 mM MgCl₂, 1 M Tris pH 7.2, 5% blocking reagent (Roche) containing 2.5 μg mL⁻¹ Cy-3-labelled telomere specific (CCCTAA) peptide nuclei acid probe (Panagene), followed by hybridisation for 2 h at room temperature in the dark. The slides were washed with 70% formamide in 2 × SSC for 2 × 15 min, followed by 2 × SSC and PBS washes for 10 min. Sections were incubated with DAPI, mounted and imaged. In depth Z stacking was used (a minimum of 40 optical slices with ×100 objective) followed by Huygens (SVI) deconvolution.

**Senescence associated beta-galactosidase staining**. Whole tissue samples were washed in PBS (pH5.5) before being fixed in 0.5% glutaraldehyde overnight and washed 2 × 15 min in PBS (pH5.5) at 4 °C. SA-β-gal activity was assessed after incubation in X-Gal solution for 90 min at 37 °C.

**Muscle morphopmetric analysis**. Mice were sacrificed at the time points described and dissected muscle was rapidly frozen in liquid nitrogen cooled isopentane to maintain structure and minimise tissue artifacts. Experimental mice and age-matched littermate controls were isolated at the same time to ensure processing was consistent between groups. Frozen muscles were equilibrated in a cryostat chamber to −20 °C and cryosections 10-μm thick were then cut from the middle third of the sample and collected on poly-L-lysine (0.5 mg/mL)–coated glass slides. Sections were allowed to air dry and were then frozen at −80 °C prior to use. Samples were brought to 4 °C on ice and fixed in a 4% w/v

0.45 mm filtered paraformaldehyde solution in 1 × PBS for 15 min at 4 °C. PFA was removed by three 5 min washes in 1 × PBS, then blocked in 10% v/v serum in 1 × PBST (0.01% Tween-20) for 1 h at RT. Primary anti-dystrophin antibody (Abcam, ab15277, 1:1000) was then applied in 1 × PBST containing 10% v/v serum for 2 h at room temperature. Three 5-min PBST washes were applied before secondary antibody conjugated to Alexa Fluor 647, with DAPI at 1:1000, incubation in PBST and 10% v/v serum for 1 h at room temperature. Sections were finally washed three times for 15 min before mounting in Vectorshield Antifade Mounting Medium (Vector Labs). Whole cross-sections of TA muscles were produced via montaged ×40 magnification tile scans (Zeiss Axio Z1 Widefield system). Morphometric analysis was performed using Fiji open source software. A macro was developed to sequentially (i) subtract background components to minimise noise that could interfere with further analysis; (ii) apply a thresholding filter for fibre border detection; (iii) generate a mask of the muscle fibre borders using the analyse particles function, simultaneously eliminating stray "non-border" signals; and (iv) overlay threshold-delimited nuclei over the border mask, before another analyse particles command was used to measure morphometric variables including "area" and "minimum Feret diameter." as previously described[46]. Simultaneous DAPI nuclear stain was used for central nucleation count. PAX7 counts were performed manually in a blinded fashion, a satellite cell was defined as having a PAX7 positive nuclei within a LAMININ cell border staining. For immunostaining the following antibodies were used anti-PAX7 PAX7 (DSHB, PAX7, 1:50), after pre-treatment with Vector Labs Mouse on Mouse Blocking Reagent (MKB-2213) according to manufacturer's instructions and anti-LAMININ (Abcam, ab11576, 1:1000).

**Bone marrow transplantation**. Atg5i mice were backcrossed 11 times to C57BL/6. At 2-month bone marrow was harvested from male Atg5i and littermate controls (containing only one allele of the two-allele system). $2 \times 10^6$ cells were transplanted into irradiated, female C57BL/6 mice (2 × 5 gy 12 h apart, transplantation occurred 24 h after the first dose). Mice were left for 1 month to enable engraftment and subsequently treated for 4 months with doxycycline at 200 ppm via their food (PicoLab Mouse Diet, 5A5X). After 4 months mice had frailty scores and blood composition analysed as described above. Additionally, to test for chimerism DNA was extracted from the blood (Qiagen DNeasy Blood & Tissue Kit) and PCRs performed to distinguish DNA from male and female origin as described previously[47]. Briefly primers amplify Kdm5c (an X-linked gene; 331 bp) and Kdm5d (a Y-linked gene; 302 bp). Whilst female mice produce only one band, males produce two, an alteration in the ratio of the upper to lower band away from that seen control DNA is indicative of altered chimerism.

Forward: 5′-CTGAAGCTTTTGGCTTTGAG-3′
Reverse: 5′-CCACTGCCAAATTCTTTGG-3′.

**Reporting summary**. Further information on research design is available in the Nature Research Reporting Summary linked to this article.

## Data and materials availability

All data and materials are available in the manuscript or upon request. Source data for Figs. 3a–c and 4b, Supplementary Figs. 1b, 6c, and 8a are provided as a source data file.

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

## Acknowledgements

We thank members of the Narita group, as well as K. Inoki of the University of Michigan, for their insights and suggestions. We are grateful to the following CRUK Cambridge Institute core facilities for advice and assistance: Histopathology, Light Microscopy (in particular H. Zecchini), PK/Bioanalytics core, and BRU. Cambridge Advanced Imaging Centre was used for EM specimen and processing facilities (in particular K.H. Muller). This work was supported by the University of Cambridge, Cancer Research UK and Hutchison Whampoa. The M.N. lab was supported by a Cancer Research UK Cambridge Institute Core Grant [C14303/A17197]. M.N. is also supported by The CRUK Early Detection Pump Priming Awards [C20/A20976] and Medical Research Council [MR/M013049/1]. C.N.J.Y. is supported by a DMU Early Career Fellowship. M.C.H.C is supported by grants from The British Heart Foundation [FS/13/3/30038], [FS/18/19/33371], and [RG/16/8/32388]. D.J. is funded by a Newcastle University Faculty of Medical Sciences Fellowship and The Academy of Medical Sciences. J.P. was supported by the BBSRC [BB/H022384/1] and [BB/K017314/1].

## Author contributions

L.D.C. and M.N. designed the research plan and interpreted the results. A.R.Y. and C.N.J.Y. isolated skeletal muscle tissue. C.N.J.Y. performed staining and analysis of muscle sections. E.J.S. and R.B. are trained pathologists and reviewed all tissue slides. E.F. and B.M.W. established and assisted with the frailty scoring. N.T.K. assisted in the interpretation of the EM that was acquired by L.D.C. and A.R.Y. K.A.W. and M.C.H.C. performed serum cytokine analyses. K.P. and M.C.H.C. performed the ANA analysis. A.L., D.J. and J.F.P. performed the TAF studies. L.D.C. and M.N. wrote the manuscript, all authors viewed and commented on.

## Competing interests

The authors declare no competing interests.
