## [Peer Review File · Nature Communications]

Reviewers' Comments:

Reviewer #1:

Remarks to the Author:

Cassidy-LD,... Narita-M, Temporal inhibition of autophagy reveals segmental reversal of aging with increased cancer risk

Submitted to Nature Communications

In this manuscript, Narita and colleagues investigate the link between autophagy inhibition and premature aging (following up, from an opposite entry point, on previous work in the field on mTOR inhibition, leading to enhanced autophagy), and its association with prolonged lifespan. Conversely, the authors find restoration of autophagy to result in partial recovery of age-associated phenotypes, but being associated with an increased rate of tumor formation. The authors conclude that the preceding inhibition of autophagy determined the elevated cancer risk (but did not test whether hyperactivation of autophagy from normal baseline level would also result in higher cancer incidence).

While the concept of this manuscript is not entirely novel, it provides – based on an elegant technical approach – important insights into autophagy-related aging mechanisms, their therapeutic implications (although one would have liked to see pharmacological data using Bafilomycin, Rapamycin and AMPK modulators to match and validate a purely gene-engineered model system in a more translational fashion) and surprising effects in terms of tumorigenesis. However, several fundamental concerns and specific questions remain before publication in Nature Communications should be considered.

Major concerns and comments

1. Potentially, the entire investigation might be flawed by a direct interference of the autophagic process with the drug metabolism of Doxycycline, either directly (ribosomal binding) or via an indirect metabolizing process (e.g. glucuronidation). Since no alternatively inducible shATG5 allele can be generated within reasonable time and effort, the authors need to provide evidence that steady-state/trough serum levels of Dox and typical direct Dox target pathways are similarly affected in wild-type vs. shATG5 mice.

Moreover, there are reports out there, saying that Dox may induce mitophagy (Xing-Y et al., *Front Cell Infect Biol* 2017), hence, the effects reported might be further specifically or non-specifically altered by autophagy-related drug effects, among them the suppression of cancer stem cell characteristics (Zhang-L et al., *Cell Cycle* 2017), which may contribute to the reversal of the liver and kidney dysfunctions (i.e. just Dox- and shAtg5-aggravated organ toxicities?) and boosted tumor incidence detected upon its withdrawal.

2. What is the nature of the widespread immune infiltrates (innate, adaptive?) as presented in Suppl. Fig. 3a? Is there evidence for autoimmune tissue damage, e.g. vasculitis? Do the LT-Atg5i mice present with higher titer autoantibodies? And how relevant is the target tissue (kidney, heart, muscle a.o.) as compared to the autophagy inhibition in hematopoietic/immune cells? Can similar phenotypes, even shortened lifespan due to accelerated aging, be observed in wild-type recipient mice being reconstituted with LT-ATG5i hematopoietic stem cells? And, upon Dox withdrawal, would those mice develop tumors others than AML and lymphoma?

3. There seem to be some controversial findings: autophagy inhibition has been shown (actually by the authors themselves) to prevent senescence (Young-AR et al., *Genes Dev* 2009), as autophagy inhibition in senescence results in secondary cell death (Dörr-JR et al., *Nature* 2013),

mTOR inhibition (promoting autophagy) extends lifespan (Harrison-DE et al., Nature 2009) and, at least in some settings, enforces senescence (Wall-M et al., Cancer Discovery 2012), albeit lowering the senescence-associated secretory phenotype (SASP) implicated in chronic inflammaging (Laberge-RM et al., Nat Cell Biol 2015; Herranz-N et al., Nat Cell Biol 2015), and continuous removal of senescent cells in a whole mammalian organism exerting a rejuvenation phenotype with reduced age-related organ pathologies (Baker-D et al., Nature 2011 and 2016), while senescence more recently was associated with cell-autonomous and non-cell-autonomous reprogramming into stemness (Mosteiro-L et al., Science 2016; Milanovic-M et al., Nature 2018), as opposed by the claim of autophagy maintaining stemness by preventing senescence (Garcia-Prat-L et al., Nature 2016).

Although it's only fair to look at the findings as they were obtained in the specific test system presented here, there is room for misinterpretation if interconnected mechanisms are not considered. Specifically: why is it necessarily "autophagy restoration" that accounts for the partial reversal of the premature aging phenotypes induced by Dox treatment before? What is the fate of shAtg5-related senescent cells: do they re-enter the cycle and divide again, or do they – upon "Atg5 restoration stress" – rather die, thereby mimicking an aging scenario, in which senescent cells are senolytically removed, as previously reported with the result of organismic rejuvenation and reduced age-related organ pathologies in an INK4a-driven senolysis transgenic mouse model (Baker-D et al., Nature 2011 and 2016)? However, the persistence of telomere-associated foci as a marker of senescent cells would argue against this view – if the comparison would just be more clear in this regard: in Fig. 4B (and following figure panels, except 4C), where is the comparison LT-Atg5i at the end of 4-mon Dox vs. two months of no Dox exposure later?

4. In part already addressed under "1.", it remains unclear whether only extracellular matrix-related alterations (such as osteopenia/kyphosis, or cardiac fibrosis) did not revert upon Dox removal due to their structural nature, or actually none of the premature aging-like phenotypes was reversed, and all improvement observed is due to lifted chronic toxicity that was exerted by Dox in the absence of Atg5.

5. Are there any characteristic biological or genetic patterns shared by the tumors emerging upon Dox withdrawal in R-Atg5i mice? For example, do they consistently present with an autophagy defect, irrespective of the interrupted Dox exposure? Or do they exhibit an inability to undergo senescence, and/or are typically driven by Ras/Braf/Mek-type oncogenes? Or is their hallmark defect massive genomic instability or aneuploidy? Unfortunately, there is virtually nothing shown in this regard.

6. Did the authors test the effect of an inducible Atg5 allele on top of a normal Atg5 gene dose at earlier or later (aging-related) time points?

Minor concerns

1. In Suppl. Fig. 2e, the authors report Congo red-positive (and birefringent?) staining of amyloid in the kidney. Is this precipitation selectively found in the glomerulus or also elsewhere throughout the body (e.g. subcutaneous fat, rectum mucosa)? What type of amyloid is it (AA, AL, or the senile type AS)? The precursor amyloid and the underlying mechanism – lack of autophagic processing of the precursor protein or indirect promotion of chronic inflammation, infection, or clonal plasma cell expansion/monoclonal gammopathy leading to amyloid formation – should be addressed.

Reviewer #2:

Remarks to the Author:

In this study, Dr. Narita's groups formulates the question as to whether decrease in autophagy

contributes to aging and whether this effect of autophagy in aging can be reverted. They have used inducible models of Atg5 deletion, a gene required for some autophagic processes, and show the deletion of Atg5 in young adult mice leads to aging phenotypes, and that restoration of the gene improves healthspan but cannot prevent the increase in spontaneous carcinogenesis.

This is an interesting and important question to address, since animals with constitutive induction of autophagy since birth have shown extension of life and health-span and lower incidence of tumors. However, an anti-aging intervention that needs to start at birth it is unlikely to be implemented. Since the most likely scenario is going to be attempts to upregulate autophagy in adult middle age individuals, this warning about the inability to prevent carcinogenesis through this intervention is important.

Overall the study is well designed and the conclusions supported, but considering that the mice die by about 6 months of age, there is some concern about how much this is really aging versus overall health conditions. Furthermore, since the phenotype that cannot be restored is mostly related with senescence, one wonders how much this is a model of aging versus a model of accelerated senescence. A later intervention (i.e. by 12 months) would have been more informative/related to aging. The other major limitation is that the authors do not present any information about autophagy. Considering the growing number of non-autophagy functions attributable to ATG genes and the already known function of Atg5 in cell death, it would be essential that they characterize the status of the autophagic system across organs.

Main points

- Repetition of some of the induction and restoration experiments at a later age (12 months) would strengthen the conclusions in relation with aging.
- Characterization of the status of the autophagic system and evidence of absence of autophagic activity needs to be presented. The only data in the whole manuscript is a reduction in LC3-II levels but considering previous reports where even in the ATG5 KO mice presence of autophagic vacuoles has been observed, they should include electron microscopy to determine whether or not there are autophagy vacuoles and measure of flux (degradation) of some typical autophagic cargo.
- Better characterization of the penetrance of the KD in different tissues should be presented. KD information is only shown for liver, heart and kidney. This data is also missing in the second hairpin studies what makes difficult to support the claims for hypomorphic reduction in Atg 5.
- In the restoration experiments, for the biochemical parameters it is very difficult to get an idea of the recovery because comparisons are done only with control (if I understand correctly in Fig. 4b the blots "R-Ctr" represent control mice but it would be important to compare with the group KD non restored too and ideally show the status of the restored also right before dox is added again. Comparing those 4 groups (or the nice time course that they did for example in the frailty studies) will be very important to understand better the surprising cancer phenotype.

Other comments:

- comments on mitochondria function/status need to be supported by functional data or soften as the authors only show levels of TOM and that is insufficient to support any functional claim
- It would be desirable including some images of the muscle as done for other tissues

-

Reviewer #3:

Remarks to the Author:

In this manuscript, Cassidy and collaborators undertake a detailed characterization of their recently generated mouse models that allow inhibition of autophagy and its reinduction, at will. Using lines for dox-inducible shRNA for Atg5, the authors demonstrate that blocking autophagy in adult mice drives several phenotypes associated to premature aging and reduced longevity, being some of them alleviated by restoration of autophagy. Importantly, the study shows that autophagy-restored mice develop tumors more frequently and earlier than autophagy-inhibited mice alone. These results are of interest to the autophagy and aging communities as autophagy has not been precisely confirmed as causal of aging-associated tissue/organ dysfunction; neither

has its reinduction been confirmed as causally involved in reversal of aging. However, several issues need further experimental evidences to strengthen their conclusions.

Specific issues:

- The authors should discuss whether premature aging in this mouse model is provoked by specific inhibition of autophagy by a multi-organ failure (provoked by inhibition of autophagy -for example, by hepatic failure) which in turn accelerates aging. This discussion should not be a show-stopper. It would just be appreciated if the authors clarify a bit more their major claims.

- Figure 2. Adult control mice (mice with normal autophagy) show some central nucleated fibers in skeletal muscle in resting state. This is not usually observed at this age. Authors should revise these data or provide a potential explanation. Some fibers in control mice are really big (bigger than normal fibers). Authors should make sure that the measurements correspond to cross-sectional areas of muscle biopsies (and not muscles cut a bit longitudinally).

- Figure 3. Is p16 expression also elevated in tissues other than liver? Telomere shortening seems quite elevated in control adult mice. Telomere/DNA damage analysis would be informative also for other tissues/organs.

- Figure 4. Could statistics be described a bit further? Could the authors also indicate more precisely and with more detail the frailty parameters that they measured? Control mice of 8 months of age seem somehow too frail for this adult age. Therefore, better understanding of this methodology will help in understanding the results of this figure.

- Figure 5. Muscle-related data of this figure is not matching always with similar data shown in Figure 2. A bit of revision will help.

- Supplem. Figure 6. Weights of all mouse models (prior/after dox treatment/removal) should be shown in this figure or elsewhere.

Reviewers' comments and point-by-point responses

Reviewer #1 (Remarks to the Author):

Cassidy-LD,... Narita-M, Temporal inhibition of autophagy reveals segmental reversal of aging with increased cancer risk

Submitted to Nature Communications

In this manuscript, Narita and colleagues investigate the link between autophagy inhibition and premature aging (following up, from an opposite entry point, on previous work in the field on mTOR inhibition, leading to enhanced autophagy), and its association with prolonged lifespan. Conversely, the authors find restoration of autophagy to result in partial recovery of age-associated phenotypes, but being associated with an increased rate of tumor formation. The authors conclude that the preceding inhibition of autophagy determined the elevated cancer risk (but did not test whether hyperactivation of autophagy from normal baseline level would also result in higher cancer incidence).

While the concept of this manuscript is not entirely novel, it provides – based on an elegant technical approach – important insights into autophagy-related aging mechanisms, their therapeutic implications (although one would have liked to see pharmacological data using Bafilomycin, Rapamycin and AMPK modulators to match and validate a purely gene-engineered model system in a more translational fashion) and surprising effects in terms of tumorigenesis. However, several fundamental concerns and specific questions remain before publication in Nature Communications should be considered.

General response to reviewer: we are grateful for the general feedback and specific recommendations by the reviewer. In particular, we believe the reviewer's suggestions to further characterise the autonomous vs non-autonomous nature of the immune phenotypes has significantly improved the manuscript. The results of this will almost certainly generate an exciting new strand of research within the lab, and strengthen existing collaborations. Point-by-point responses to the reviewer's comments can be found below.

Major concerns and comments

1. Potentially, the entire investigation might be flawed by a direct interference of the autophagic process with the drug metabolism of Doxycycline, either directly (ribosomal binding) or via an indirect metabolizing process (e.g. glucuronidation). Since no alternatively inducible shATG5 allele can be generated within reasonable time and effort, the authors need to provide evidence that steady-state/trough serum levels of Dox and typical direct Dox target pathways are similarly affected in wild-type vs. shATG5 mice.

Moreover, there are reports out there, saying that Dox may induce mitophagy (Xing-Y et al., Front Cell Infect Biol 2017), hence, the effects reported might be further specifically or non-specifically altered by autophagy-related drug effects, among them the suppression of cancer stem cell characteristics (Zhang-L et al., Cell Cycle 2017), which may contribute to the reversal of the liver and kidney dysfunctions (i.e. just Dox- and shAtg5-aggravated organ toxicities?) and boosted tumor incidence detected upon its withdrawal.

We have attempted to address the reviewers concern here in two ways. Firstly, as suggested we took serum doxycycline levels from control and experimental cohorts and measured their serum steady-state levels after 4 months of treatment. We found no difference between the mice based on genotype and the values are comparable to those previously published [PMID 27423155]. These data

can now be seen in new Supplementary Figure 1d. Of note, in this study, we used a low dose regimen of dox (200PPM) whilst in comparison many labs use 625PPM, often results in serum levels up to 10X times higher than our mice display. We are able to do this due to the high sensitivity afforded by the Cags-rtTA3 system. However, we accept that it is impossible to rule out synergistic effects of the dox and autophagy inhibition (as it would equally be for other widely used inducible model systems i.e. tamoxifen). As such we have also added this as a discussion point (3rd paragraph of the discussion) to highlight what is undoubtedly an important caveat, and one that to our knowledge has never been rigorously addressed in similar genetically engineered inducible mouse model systems.

2. What is the nature of the widespread immune infiltrates (innate, adaptive?) as presented in Suppl. Fig. 3a?

Is there evidence for autoimmune tissue damage, e.g. vasculitis? Do the LT-Atg5i mice present with higher titer autoantibodies?

And how relevant is the target tissue (kidney, heart, muscle a.o.) as compared to the autophagy inhibition in hematopoietic/immune cells?

Can similar phenotypes, even shortened lifespan due to accelerated aging, be observed in wild-type recipient mice being reconstituted with LT-ATG5i hematopoietic stem cells?

And, upon Dox withdrawal, would those mice develop tumors others than AML and lymphoma?

The reviewer raises a number of very interesting questions and we have attempted to address as much as possible here. Firstly, representative staining and quantification for immune infiltrations in various tissues can be seen in new Supplementary Figure 5b-d.

In terms of autoimmunity, we have not seen any reproducible evidence of vasculitis via analysis with a pathologist. However, as suggested by the reviewer, we have also performed an Antinuclear Antibody Test to determine if there is any evidence of autoimmunity in the Atg5i mice. The results can be found in new Supplemental Figure 5e and suggest that there is an increase in the frequency of autoimmunity in the Atg5i mice. A positive signal was determined to be present in 5/12 (41.7%) Atg5i mice in comparison to 1/6 (16.7%) of age matched control mice, with the predominant pattern homogenous and/or speckled. Of note, while autoimmunity was only positive in 5/12 Atg5i mice, immune expansion and infiltration is a universal phenotype.

We agree with the reviewer that the relevance of the immune system autophagy loss versus the tissue autophagy loss is of great interest. As such we have performed the bone marrow transplant (BMT) experiment as suggested. To do this, we took Control and Atg5i mice that had recently been backcrossed 11 times to C57/BL6, and reconstituted 5 lethally irradiated wild-type C57/BL6 mice per genotype (N= 5 Ctrl and 5 Atg5i). After 1 month, these mice were treated with doxycycline and at 4 months blood was taken for analysis. The results are interesting. Previously, we noted an expansion of the myeloid lineage and increased cellularity of the peripheral immune system in Atg5i mice. However, in our wild-type mice with Atg5i bone marrow, we only saw a myeloid skewing (new Fig. 2e), but this did not coincide with an expansion of cellularity (peripheral white blood cell count, new Fig. 2b), thus providing a separation of immune phenotypes. This suggests that the loss of autophagy in the immune cells produces a myeloid bias (cell-autonomous effect), whilst loss of autophagy in non-immune cells stimulates their expansion (non-cell-autonomous effect). Furthermore, PCR based analysis suggests that bone marrow from Atg5i mice actually constituted a lower fraction of the peripheral blood.

Upon closer inspection, our results are highly consistent with the published literature. To place this experiment in context, an analogous experiment was performed by the lab of Emmanuelle Passegue (PMID 28241143). Here the authors knocked-out a key autophagy gene (Atg12) in the immune system using the Mx1-Cre model (using pIC to induce recombination), and note an expansion of the immune

system and a myeloid skewing, an effect recapitulated in our Atg5i mice. Next, they transplanted bone marrow from Atg12^{flox/flox} mice (not yet treated with pIC and as such autophagy competent) into wild-type hosts and left them 2 months to enable reconstitution. After treatment with pIC, which induced deletion of Atg12, instead of an expansion of the peripheral immune system, the authors note a gradual reduction in chimerism, but these mice still recapitulated the myeloid skewing. As such the data are in agreement with our new Atg5i-BMT data. It is important to note that, the Mx1-Cre model used by Passegue and colleagues, also induces liver recombination, as such they do not use a pure immunological system and we would speculate that their immune expansion may be in-part, driven by the resultant liver autophagy loss.

Finally, while we fully agree that expanding these BMT experiments as well as performing longevity and tumour studies would be of great interest, these are large studies spanning three to five years. We have however performed frailty scoring of mice with BMT uncovered no evidence of accelerated ageing (new Supplementary Figure 5f).

We would like to reiterate that these were excellent points raised by the reviewer and we thank them for their suggestion. These results are undoubtedly very interesting and will form the basis of new avenues of work.

3. There seem to be some controversial findings: autophagy inhibition has been shown (actually by the authors themselves) to prevent senescence (Young-AR et al., Genes Dev 2009), as autophagy inhibition in senescence results in secondary cell death (Dörr-JR et al., Nature 2013), mTOR inhibition (promoting autophagy) extends lifespan (Harrison-DE et al., Nature 2009) and, at least in some settings, enforces senescence (Wall-M et al., Cancer Discovery 2012), albeit lowering the senescence-associated secretory phenotype (SASP) implicated in chronic inflammaging (Laberge-RM et al., Nat Cell Biol 2015; Herranz-N et al., Nat Cell Biol 2015), and continuous removal of senescent cells in a whole mammalian organism exerting a rejuvenation phenotype with reduced age-related organ pathologies (Baker-D et al., Nature 2011 and 2016), while senescence more recently was associated with cell-autonomous and non-cell-autonomous reprogramming into stemness (Mosteiro-L et al., Science 2016; Milanovic-M et al., Nature 2018), as opposed by the claim of autophagy maintaining stemness by preventing senescence (Garcia-Prat-L et al., Nature 2016).

Although it's only fair to look at the findings as they were obtained in the specific test system presented here, there is room for misinterpretation if interconnected mechanisms are not considered. Specifically: why is it necessarily "autophagy restoration" that accounts for the partial reversal of the premature aging phenotypes induced by Dox treatment before? What is the fate of shAtg5-related senescent cells: do they re-enter the cycle and divide again, or do they – upon "Atg5 restoration stress" – rather die, thereby mimicking an aging scenario, in which senescent cells are senolytically removed, as previously reported with the result of organismic rejuvenation and reduced age-related organ pathologies in an INK4a-driven senolysis transgenic mouse model (Baker-D et al., Nature 2011 and 2016)? However, the persistence of telomere-associated foci as a marker of senescent cells would argue against this view – if the comparison would just be more clear in this regard: in Fig. 4B (and following figure panels, except 4C), where is the comparison LT-Atg5i at the end of 4-mon Dox vs. two months of no Dox exposure later?

It is true that the relationship between autophagy and senescence is complex and highly context dependent. It is important to note that both basal autophagy and stress-induced autophagy are relevant for senescence. We have shown that stress- (genotoxic and oncogenic) induced autophagy contributes the inflammatory SASP, although its impact on senescence 'arrest' is marginal. In contrast, it is known that basal autophagy is critical for cellular fitness thus depletion of basal autophagy would promote senescence. In addition, increasing evidence indicates that autophagy has multiple action

points in senescence effectors (the SASP, cGAS-STNG etc.). We have extensively discussed this point in our recent review article in *Genes & Dev* (PMID 30709901).

As suggested by the reviewer, we have now included new data for the comparison of mice at 4 months and 4 months post dox removal in new Figure 4b. In the liver, we find that p16 levels (whilst elevated in comparison to controls in the restored cohorts) are lower than in the cohorts with autophagy inhibited. Interestingly, the kidney displays only a modest increase in p16 during autophagy inhibition, and is completely recovered in the restored cohorts; suggestive of differential tissue susceptibility to autophagy inhibition. We agree that this is an important comparison for relative data (such as western blots) as it enables the direct comparison with the 4 month-timepoint for the senescent and autophagy markers. While how the (partial) reduction of senescence load was achieved (either senescence escape, cell death, or immune elimination), it is possible that the reduced senescence-load might contribute to the partial age-reverse upon autophagy restoration. However, this is too speculative and we did not mention this possibility in discussion. This is a fundamental question that we hope to address in the future.

The corresponding 4-month data for white blood cell counts (old Figure 4d) and serum cytokine levels (old Figure 4e), can be seen in Figure 2b and 2d, respectively. For the 4 month RDW values we have now included these in the manuscript (Figure 4f). Briefly, there is no RDW differences at the 4-month timepoint and, whilst there appears an age-related increase at the 12 months, a further increase can be seen in the R-Atg5i cohorts in comparison to the R-Ctrl. We thank the reviewer for the suggestion to include these data.

4. In part already addressed under “1.”, it remains unclear whether only extracellular matrix-related alterations (such as osteopenia/kyphosis, or cardiac fibrosis) did not revert upon Dox removal due to their structural nature, or actually none of the premature aging-like phenotypes was reversed, and all improvement observed is due to lifted chronic toxicity that was exerted by Dox in the absence of Atg5. Please see “1”

5. Are there any characteristic biological or genetic patterns shared by the tumors emerging upon Dox withdrawal in R-Atg5i mice? For example, do they consistently present with an autophagy defect, irrespective of the interrupted Dox exposure? Or do they exhibit an inability to undergo senescence, and/or are typically driven by Ras/Braf/Mek-type oncogenes? Or is their hallmark defect massive genomic instability or aneuploidy? Unfortunately, there is virtually nothing shown in this regard.

The reviewer raises many excellent questions here. We agree that addressing the role of autophagy in these tumours is an important and performed IHC for p62 (a marker associated with a block in autophagy). An example can be seen in Figure 6d. We have seen no evidence that autophagy continues to be inhibited in the tumours from R-Atg5i mice and have added a line to the last paragraph in the results section to convey this message. Further, systematic characterisation of tumours is currently ongoing, particularly focusing on genome-sequence approaches. We emphasise that our mice do not just model premature ageing but also age-associated spontaneous tumorigenesis. As implied by the reviewer, it is conceivable that genomic instability that might have been induced by autophagy defect, could increase the cancer risk, which causes cancer development only in the autophagy proficient condition. Our model raises a possibility that we might even be able to observe (epi)genomic events that precede cancer development. We are actively working on this fundamental question.

6. Did the authors test the effect of an inducible Atg5 allele on top of a normal Atg5 gene dose at earlier or later (aging-related) time points?

Unfortunately, we did not develop a mouse with an inducible Atg5 allele. However, a GEMM study has been published wherein Atg5 is constitutively overexpressed from embryogenesis (PMID: 23939249) and recently another paper was published describing a GEMM with constitutively high basal autophagy flux (PMID:29849149). Both studies argue that increased autophagy leads to an extension of health and life-span. Both of these studies were cited in the original manuscript (introduction and discussion) as they are complementary to our data. Their data combined with ours suggest that maintaining a high basal autophagic flux can help prevent damage that normally accumulates with age, however our data additionally provide evidence that 1) reduction of autophagy (in adults) promotes ageing, 2) such ageing is partially reversible, and 3) autophagy decline is not sufficient for tumorigenesis but increases a cancer risk.

We argue this knowledge is essential as the therapeutic promotion of autophagy in the general population would not occur from birth, it would instead occur later in life. Perhaps after irreversible damage has already accumulated.

Minor concerns

1. In Suppl. Fig. 2e, the authors report Congo red-positive (and birefringent?) staining of amyloid in the kidney. Is this precipitation selectively found in the glomerulus or also elsewhere throughout the body (e.g. subcutaneous fat, rectum mucosa)? What type of amyloid is it (AA, AL, or the senile type AS)? The precursor amyloid and the underlying mechanism – lack of autophagic processing of the precursor protein or indirect promotion of chronic inflammation, infection, or clonal plasma cell expansion/monoclonal gammopathy leading to amyloid formation – should be addressed.

As this data was ancillary we have removed this data and discussion points. The reviewer quite rightly points out this is fairly preliminary and we believe a more significant, independent study would be required beyond our current capacity.

Reviewer #2 (Remarks to the Author):

In this study, Dr. Narita's groups formulates the question as to whether decrease in autophagy contributes to aging and whether this effect of autophagy in aging can be reverted. They have used inducible models of Atg5 deletion, a gene required for some autophagic processes, and show the deletion of Atg5 in young adult mice leads to aging phenotypes, and that restoration of the gene improves healthspan but cannot prevent the increase in spontaneous carcinogenesis. This is an interesting and important question to address, since animals with constitutive induction of autophagy since birth have shown extension of life and health-span and lower incidence of tumors. However, an anti-aging intervention that needs to start at birth it is unlikely to be implemented. Since the most likely scenario is going to be attempts to upregulate autophagy in adult middle age individuals, this warning about the inability to prevent carcinogenesis through this intervention is important.

Overall the study is well designed and the conclusions supported, but considering that the mice die by about 6 months of age, there is some concern about how much this is really aging versus overall health conditions. Furthermore, since the phenotype that cannot be restored is mostly related with senescence, one wonders how much this is a model of aging versus a model of accelerated senescence. A later intervention (i.e. by 12 months) would have been more informative/related to aging. The other major limitation is that the authors do not present any information about autophagy. Considering the growing number of non-autophagy functions attributable to ATG genes and the already known

function of Atg5 in cell death, it would be essential that they characterize the status of the autophagic system across organs.

General response to reviewer: We appreciate the constructive and positive comments from the reviewer regarding both the design and importance of the work. We have now provided substantially more evidence to the manuscript, including EM, an *in vivo* autophagy flux assay, and additional staining across various tissues to reinforce the loss of autophagy across multiple organs. This combined with our technical paper describing the model in *Autophagy* (PMID 29999454) provides strong supportive evidence of autophagy perturbation across the majority of tissues. A point-by-point responses to the specific reviewer's comments can be found below.

Main points

- Repetition of some of the induction and restoration experiments at a later age (12 months) would strengthen the conclusions in relation with aging.

Whilst performing the experiments in older cohorts would provide an interesting comparison (as we predict phenotypes may become more pronounced and recovery would be less robust) we do not believe they are necessary in this instance to support the notion that Atg5i mice have accelerated ageing, as we expect the phenotypes to be broadly similar.

- Characterization of the status of the autophagic system and evidence of absence of autophagic activity needs to be presented. The only data in the whole manuscript is a reduction in LC3-II levels but considering previous reports where even in the ATG5 KO mice presence of autophagic vacuoles has been observed, they should include electron microscopy to determine whether or not there are autophagy vacuoles and measure of flux (degradation) of some typical autophagic cargo.

To address the reviewer's comments, we have performed EM on liver tissue from Atg5i and control mice that have been treated with doxycycline for 6 weeks. Of note, we can see the appearance of stacked and vacuolated membranes that one would expect to see upon autophagy inhibition. This data is now included in new Supplementary Figure 1a.

Additionally, we have performed an *in vivo* flux experiment wherein mice had autophagy inhibited for 3 weeks before dox was removed and a time-course was followed to watch the re-establishment of autophagy with the subsequent removal of typical autophagic cargo that had built up over time (new Supplementary Figure 1b and c). Note, the timing of Atg5 recovery and reduction of accumulated cargos (p62, NBR1) are well correlated: the former slightly precedes the latter, adding further evidence to support Atg5 and autophagy were inhibited.

We would also like to re-iterate that while these mice have reduced autophagy (phenotypically supported through p62/Sqstm1, LC3-I build-up, and by EM), as it is fundamentally a knockdown experiment (more physiological than KO experiments), we do not preclude the possibility that some autophagosome biogenesis is occurring and that flux through the autophagy pathway may be present. We have furthermore added a line to the second paragraph of the discussion as this is a point of central importance, particularly when comparing this model to the knockout models. We thank the reviewer for highlighting this and believe the discussion is now more comprehensive in this regard.

- Better characterization of the penetrance of the KD in different tissues should be presented. KD information is only shown for liver, heart and kidney. This data is also missing in the second hairpin studies what makes difficult to support the claims for hypomorphic reduction in Atg5.

In 2018, we published a technical paper (PMID 29999454) extensively describing the mouse model and including the penetrance of the knockdown across a broad range of tissues which included pancreas, spleen, skeletal muscle, liver, lung, heart, seminal vesicles and brain, as well as using MEFs in *in vitro* experiments. To further validate the model, we have now included example images of Sqstm1/p62 accumulation across several tissues (spleen, lung, salivary gland, pancreas, skeletal muscle, heart) in new Supplementary Figure 2a, in addition to those data already present (Figure 3, 4, 5 and Supplementary Figure 6).

We have also added new heart and lung IHC for Sqstm1/p62 and LC3 for the Atg5i_2 mouse model in new Supplementary Figure 7f. Additionally, the hypomorphic nature of the second hairpin mouse is based on three findings. The first is that they do not display some of the key phenotypes associated Atg5 KO, the first hairpin mouse does. The second is that the mice do not display as strong a level of p62 build-up or diffuse LC3 build up. The third is that the mice display similar ageing phenotypes but at a reduced rate. We chose the second shRNA sequence from our initial screening, in which this exhibited the second highest efficiency of KD, shAtg5 #3 (Atg5_1654) in our paper (PMID 29999454), the best one was used for the first Atg5i mouse. Additionally, we have now included better description of the second shRNA in the methods section. Nevertheless, both hairpins display a highly efficient KD via western blot analysis, and that the difference between the two hairpins is very low. However, it is known that inhibiting Atg5 dependent autophagy requires a very high KD efficiency (PMID 16647067), explaining that such a small difference in the Atg5 level results in substantial phenotypical differences.

- In the restoration experiments, for the biochemical parameters it is very difficult to get an idea of the recovery because comparisons are done only with control (if I understand correctly in Fig. 4b the blots "R-Ctr" represent control mice but it would be important to compare with the group KD non-restored too and ideally show the status of the restored also right before dox is added again. Comparing those 4 groups (or the nice time course that they did for example in the frailty studies) will be very important to understand better the surprising cancer phenotype.

We thank the reviewer for this suggestion, we agree that comparing to the 4-month timepoint in the same blot is highly informative. We have now updated this figure as suggested and it is possible to see the restoration of Atg5 and LC3 (-I and -II) to similar basal levels for Liver and kidney tissue (new Fig 4b). Of note this result also highlights how senescence markers alter over time in different tissues. Consistent with telomere-associated γ -H2AX foci (TAF) data, in liver p16 levels remained high in R-Atg5i (compared to littermate controls), although this is not as prominent as in the LT-Atg5i mice (4-months dox), suggesting incomplete recovery after autophagy restoration. Meanwhile in the kidney, p16 exhibits only modest increase while autophagy is inhibited, and this appears to fully recover 4 months after dox removal.

Other comments:

- comments on mitochondria function/status need to be supported by functional data or soften as the authors only show levels of TOM and that is insufficient to support any functional claim

As correctly pointed out by the reviewer, we have not performed any functional work, as such we have altered our text to reflect this.

- It would be desirable including some images of the muscle as done for other tissues

We have now added images of the muscle, both in regards to p62 and LC3 in new supplementary figure 2, as well as example images for our morphometry analysis in Supplementary Figure 6a

Reviewer #3 (Remarks to the Author):

In this manuscript, Cassidy and collaborators undertake a detailed characterization of their recently generated mouse models that allow inhibition of autophagy and its reinduction, at will. Using lines for dox-inducible shRNA for Atg5, the authors demonstrate that blocking autophagy in adult mice drives several phenotypes associated to premature aging and reduced longevity, being some of them alleviated by restoration of autophagy. Importantly, the study shows that autophagy-restored mice develop tumors more frequently and earlier than autophagy-inhibited mice alone. These results are of interest to the autophagy and aging communities as autophagy has not been precisely confirmed as causal of aging-associated tissue/organ dysfunction; neither has its reinduction been confirmed as causally involved in reversal of aging. However, several issues need further experimental evidences to strengthen their conclusions.

General response to reviewer: We thank the reviewer not only for their positive comments regarding the importance of the work, and the subsequent interest to the community, but also for his/her constructive comments on the muscle related datasets. We have now re-isolated and performed all the assays with particular care taken in the isolation, fixation, cutting depth, and image analysis for all the samples. As such we now believe the muscle dataset is far more robust than before. In addition, we have also added a new dataset (4 months on dox) to the manuscript. Point-by-point responses to the specific reviewer's comments can be found below.

Specific issues:

- The authors should discuss whether premature aging in this mouse model is provoked by specific inhibition of autophagy by a multi-organ failure (provoked by inhibition of autophagy -for example, by hepatic failure) which in turn accelerates aging. This discussion should not be a show-stopper. It would just be appreciated if the authors clarify a bit more their major claims.

The reviewer raises a very important discussion point that we agree should be expanded upon further. While it is possible that organ damage might contribute to the accelerated ageing phenotypes, the premature ageing phenotypes of our mice are not driven specifically by these organ damages (e.g. the hepatomegaly phenotypes), as these are absent from LT-Atg5i₂ mice. This point has been reinforced in the Discussion.

- Figure 2. Adult control mice (mice with normal autophagy) show some central nucleated fibers in skeletal muscle in resting state. This is not usually observed at this age. Authors should revise these data or provide a potential explanation. Some fibers in control mice are really big (bigger than normal fibers). Authors should make sure that the measurements correspond to cross-sectional areas of muscle biopsies (and not muscles cut a bit longitudinally).

- Figure 5. Muscle-related data of this figure is not matching always with similar data shown in Figure 2. A bit of revision will help.

We thank the reviewer for their suggestions here. To address these two issues (Figure 2 and Figure 5), we re-isolated tissues from mice with particular care taken with the snap freezing technique and subsequent sectioning. In addition, in all tissues we sectioned 1/3 into the tissue in an attempt to get a similar depth for analysis. For all analysis, we next manually removed all fibres that appeared to be cut longitudinally in a blinded fashion. Due to this we believe the data are now far more robust and comparable.

- Figure 3. Is p16 expression also elevated in tissues other than liver? Telomere shortening seems quite elevated in control adult mice. Telomere/DNA damage analysis would be informative also for other tissues/organs.

We also showed western blotting for p16 in the muscle in the original version (now in new Supplementary Figure 6c). We have now performed p16 western blotting for kidneys (new Figure 4b), in addition to expanding our TAF counting to lung and heart also which displayed the same general trend as in the liver (new Supplemental Figure 6d and e).

For the TAF analysis, the numbers are similar to those previously reported by Hewitt et al. (PMID 22426229, Figure 5). Importantly we should add that TAF do not necessarily represent shortened telomeres, a concept described and discussed in this original paper, just DNA damage at telomeres (mostly irreparable). We have added a line into to emphasise this for clarification.

- Figure 4. Could statistics be described a bit further? Could the authors also indicate more precisely and with more detail the frailty parameters that they measured? Control mice of 8 months of age seem somehow too frail for this adult age. Therefore, better understanding of this methodology will help in understanding the results of this figure.

We have expanded this section in the methods and fully agree with the reviewer that more detailed information would be beneficial, especially considering a number of frailty scoring systems exist in the literature. This can now be seen in the updated methods section of the manuscript. Our scores are similar both to the original published paper, and an independently scored aged cohort at a different location (University of Newcastle, E. Fielder, personal communication). As the frailty score is in essence a qualitative score, it is open to differences in the scoring between individuals and research groups. To reduce variability and correctly score these qualitative metrics we believe it is essential therefore to use the same researchers to measure the same mice (in a blinded fashion), as we have done in this instance, as this provides a degree of internal control.

- Supplem. Figure 6. Weights of all mouse models (prior/after dox treatment/removal) should be shown in this figure or elsewhere.

We have now added as requested as new Supplemental Figure 9a-b.

Reviewers' Comments:

Reviewer #1:

Remarks to the Author:

Cassidy-LD,... ...Narita-M, Temporal inhibition of autophagy reveals segmental reversal of aging with increased cancer risk

Submitted to Nature Communications

This is now a substantial revision of the scientifically very interesting manuscript on the link between autophagy inhibition and premature aging with a specific focus on autophagy restoration that led to partial recovery of age-associated phenotypes, but an increased rate of tumor formation. Despite an important extension of previous work in the field and the elegant investigation of a very useful conditional ATG5 mouse model, the initially submitted version raised a number of fundamental concerns and specific questions.

Being aware of the many global and detailed aspects I pointed out in my statement, I would like to thank the authors for an unusually careful and comprehensive discussion of the critique, and the significant additional experimental work now provided in the re-submission. Although not all concerns were addressed (with some of them, I admit, hard-to-impossible to solve in reasonable time with reasonable efforts), I have no further objections. The current manuscript is now much stronger, more compelling and definitely, from my perspective, deserves publication in Nature Communications.

Reviewer #3:

Remarks to the Author:

Main points

-Repetition of some of the induction and restoration experiments at a later age (12 months) would strengthen the conclusions in relation with aging.

I agree with the authors that doing these new experiments (at 12 months of age) would be informative; yet, it is clear that Atg5i mice are a new model of accelerated aging, and at this point there is no need to do the extra effort of using older mice.

-Characterization of the status of the autophagic system and evidence of absence of autophagic activity needs to be presented.

The authors present sufficient new data on autophagy analysis (EM, flux assays, p62/Sqstm1, LC3-I build-up).

- Better characterization of the penetrance of the KD in different tissues should be presented.

The authors already showed data related to KD penetrance in a previous paper, and in the present manuscript they add heart and lung tissues and analyze different autophagy parameters.

- Restoration/recovery experiments.

The figure is now updated and the restoration of Atg5 and LC3 to similar basal levels for liver and kidney is clear. Recovery is observed in different tissues and to different extent.

Minor points are well addressed.

In sum, I think that the authors have addressed well the major concerns of Reviewer 2.